# Antagonism in olfactory receptor neurons and its implications for the perception of odor mixtures

Gautam Reddy[1†], Joseph D Zak[2,3†], Massimo Vergassola[1]*, Venkatesh N Murthy[2,3]*

[1]Department of Physics, University of California, San Diego, La Jolla, United States; [2]Department of Molecular Cellular Biology, Harvard University, Cambridge, United States; [3]Center for Brain Science, Harvard University, Cambridge, United States

**Abstract** Natural environments feature mixtures of odorants of diverse quantities, qualities and complexities. Olfactory receptor neurons (ORNs) are the first layer in the sensory pathway and transmit the olfactory signal to higher regions of the brain. Yet, the response of ORNs to mixtures is strongly non-additive, and exhibits antagonistic interactions among odorants. Here, we model the processing of mixtures by mammalian ORNs, focusing on the role of inhibitory mechanisms. We show how antagonism leads to an effective 'normalization' of the ensemble ORN response, that is, the distribution of responses of the ORN population induced by any mixture is largely independent of the number of components in the mixture. This property arises from a novel mechanism involving the distinct statistical properties of receptor binding and activation, without any recurrent neuronal circuitry. Normalization allows our encoding model to outperform non-interacting models in odor discrimination tasks, leads to experimentally testable predictions and explains several psychophysical experiments in humans.

DOI: https://doi.org/10.7554/eLife.34958.001

*For correspondence:
massimo@physics.ucsd.edu (MV);
vnmurthy@fas.harvard.edu (VNM)

†These authors contributed equally to this work

Competing interests: The authors declare that no competing interests exist.

## Introduction

The olfactory system, like other sensory modalities, is entrusted to perform certain basic computational tasks. Of primary importance is the specific identification of odors and the recognition of isolated sources or objects in an olfactory scene. A typical scene in a natural environment is complex: the olfactory landscape is determined by the chemical composition of odorants released by the objects, the stoichiometry of the mixture and the physical location of the objects relative to the observer. An efficient olfactory system is expected to eliminate irrelevant background components and de-mix contextually relevant components received as a blend (*Ache et al., 2016*; *Cardé and Willis, 2008*; *Gottfried, 2010*; *Hopfield, 1999*; *Howard and Gottfried, 2014*; *Jinks and Laing, 1999*; *Knudsen et al., 1993*; *Pentzek et al., 2007*; *Raguso, 2008*; *Riffell, 2012*; *Riffell et al., 2008*; *Riffell et al., 2014*; *Rokni et al., 2014*; *Stevenson and Wilson, 2007*; *Szyszka and Stierle, 2014*; *Thomas-Danguin et al., 2014*).

The importance of filtering a complex background is shared by the olfactory and the adaptive immune systems. In the latter, lymphocytes must quickly and accurately identify a small fraction of foreign ligands in a sea of native ligands (*Abbas et al., 2014*). Inhibitory feedback plays a key role in meeting the challenge of a proper combination of rapidity, sensitivity and specificity (*François et al., 2013*). Inhibitory interactions in the form of receptor antagonism have indeed been observed in experiments with olfactory receptor neurons (ORNs) (*Oka et al., 2004*; *Takeuchi et al., 2009*; *Kurahashi et al., 1994*), although it has not been quantified systematically. For instance, the response of cells expressing the mOR-EG receptor is strongly suppressed when methyl isoeugenol is

**eLife digest** When ordering in a coffee shop, you probably recognize and enjoy the aroma of freshly roasted coffee beans. But as well as coffee, you can also smell the croissants behind the counter and maybe even the perfume or cologne of the person next to you. Each of these scents consists of a collection of chemicals, or odorants. To distinguish between the aroma of coffee and that of croissants, your brain must group the odorants appropriately and then keep the groups separate from each other.

This is not a trivial task. Odorants bind to proteins called odorant receptors found on the surface of cells in the nose called olfactory receptor neurons. But each odorant does not have its own dedicated receptor. Instead, a single odorant will bind to multiple types of odorant receptors, and thus, each olfactory receptor neuron may respond to multiple odorants. So how does the brain encode mixtures of odorants in a way that allows us to distinguish one aroma from another?

Reddy, Zak et al. have developed a computational model to explain how this process works. The model assumes that an odorant triggers a response in an olfactory receptor neuron via two steps. First, the odorant binds to an odorant receptor. Second, the bound odorant activates the receptor. But the odorant that binds most strongly to a receptor will not necessarily be the odorant that is best at activating that receptor.

This allows a phenomenon called competitive antagonism to occur. This is when one odorant in a mixture binds more strongly to a receptor than the other odorants, but only weakly activates that receptor. In so doing, the strongly bound odorant prevents the other odorants from binding to and activating the receptor. This helps tame the dominating influence of background odors, which might otherwise saturate the responses of individual olfactory receptor neurons.

Reddy, Zak et al. show that processes such as competitive antagonism enable olfactory receptor neurons to encode all of the odors within a mixture. The model can explain various phenomena observed in experiments and it adds to our understanding of how the brain generates our sense of smell. The model may also be relevant to other biological systems that must filter weak signals from a dominant background. These include the immune system, which must distinguish a small set of foreign proteins from the much larger number of proteins that make up our bodies.
DOI: https://doi.org/10.7554/eLife.34958.002

delivered together with the receptor's cognate ligand, eugenol, at equal concentrations (*Oka et al., 2004*). Further evidence of intensity suppression and overshadowing (i.e. when one odorant makes another indiscernible) in the perception of odorant mixtures comes from psychophysical observations (*Lawless, 1997*; *Keller and Vosshall, 2004*; *Doty and Laing, 2015*; *Thomas-Danguin et al., 2014*). The importance of peripheral interactions in shaping mixture perception has been directly shown by electrophysiological and psychophysical measurements (*Bell et al., 1987*; *Laing and Willcox, 1987*; *Chaput et al., 2012*). However, the functional role, if any, of inhibition at the ORN level remains unknown.

Each ORN expresses receptors of a particular type, which typically display broad sensitivities to different odorants, whereas each odorant binds promiscuously to receptors of many types. The axons of ORNs of a common type converge onto glomeruli, where the axon terminals form synaptic contacts with mitral and tufted (M/T) projection neurons leading to the cortex, as well as local periglomerular (PG) interneurons. The activation of an individual glomerulus therefore represents the activation of a single ORN type. Discriminatory computations are carried out by brain regions such as the olfactory cortex, which receive combinatorial information from the entire ORN ensemble. To achieve a quantitative description of ORN inhibitory effects, it is then imperative to take their global nature into account. In other words, it is necessary to address the knowledge gap between the mixture response properties of a single ORN, the ensemble glomerular response, and ultimately its influence on odor discrimination and perception, which constitutes the goal of the present work.

Previous computational models that examined discrimination tasks have, for simplicity, assumed a linear summation model of mixture response at the ORN level (*Hopfield, 1999*; *Koulakov et al., 2007*; *Zhang and Sharpee, 2016*; *Zwicker et al., 2016*; *Mathis et al., 2016*; *Grabska-Barwińska et al., 2017*). Conversely, our emphasis is on explicitly characterizing the ORNs'

biophysical attributes, with a focus on mixture response properties. A key aspect of the model is that odorant-receptor interactions depend on two distinct features: the sensitivity to binding and the efficiency of activation after binding, respectively. Competitive antagonism occurs when a component in a mixture that binds strongly, activates the downstream transduction pathway less effectively compared to other components. While this might naively seem disadvantageous, we show how an antagonistic encoding model has inherent normalization properties, leading to superior performance in odor discrimination and identification tasks. Finally, we make an explicit connection between a variety of psychophysical observations related to the perception of odorant mixtures and inhibitory effects at the single ORN level, providing a potential neurobiological basis to perceptual phenomena.

## Results

### Biophysics of mammalian olfactory receptor neurons

Odorants in the nasal cavity are captured by G-protein-coupled-receptors located on the cilia of ORNs. The conversion of chemical binding events into transduction currents in the cilia leads to spike signals transmitted to the brain (see Refs. [*Pifferi et al., 2010*; *Kleene, 2008*] for reviews). The signal transduction pathway is complex; here, we build a chemical rate model of the mammalian ORN meant to capture its major response properties. The model complements and extends previous work on mixture interactions (*Rospars et al., 2008*; *Cruz and Lowe, 2013*). In *Figure 1A*, we illustrate the transduction pathway as modeled, and present a summarized version below (see Materials and methods for details).

The binding of an odorant to a receptor induces the activation of the odor-receptor complex via a two-step process. For an odorant $X$, its interactions at the receptor level are then represented by a two-step process:

$$R + X \underset{}{\overset{\kappa_1}{\rightleftharpoons}} RX \underset{}{\overset{\kappa_2}{\rightleftharpoons}} RX^*, \tag{1}$$

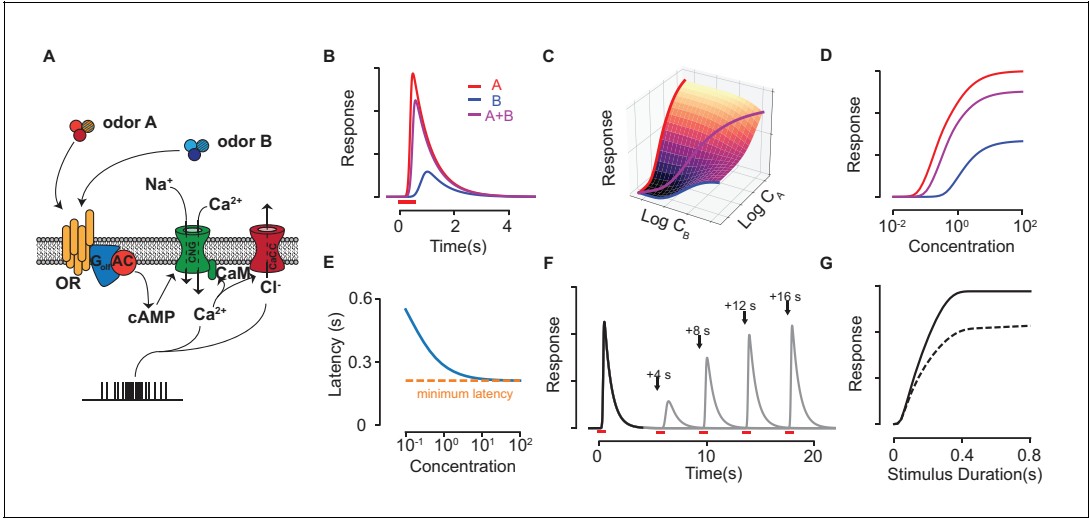

**Figure 1.** Response properties of an ORN in our biophysical model. (A) Scheme of the modeled ORN signal transduction pathway. (B) The temporal responses of an ORN to odorants $A$ (red) and $B$ (blue) delivered separately and as a mixture (magenta). (C–D) The peak response of the mixture for different concentrations of $A$ and $B$. The three colored curves in (C) are plotted separately in (D) on a single axis, whose scale corresponds to $C_A$, $C_B$ and $C_A + C_B$ for $A$, $B$ and the mixture $A + B$, respectively. (E) The response latency *vs* odorant concentration. The red, dashed line shows the minimum possible latency due to the limiting receptor activation and cAMP production steps in the signal transduction, which varies between a few tens of milliseconds to a few hundred milliseconds depending on the particular odorant-receptor pair under consideration (*Ghatpande and Reisert, 2011*; *Rospars et al., 2003*). (F–G) The $Ca^{2+}$-based adaptation properties of the biophysical model. In (F) the first odorant pulse at $t = 0$ is followed by a second pulse at each of the four shown times in separate trials. Full response is recovered after a few tens of seconds. (G) The peak response of a pulse (solid) and a second pulse (dashed) delivered 10 s later against the pulse duration of both pulses.
DOI: https://doi.org/10.7554/eLife.34958.003

where $\kappa_1$ and $\kappa_2$ are the ratios of backward to forward rates for the binding and the activation steps. R, RX and RX$^*$ represent unbound, bound inactive and bound activated receptors, respectively. Activated complexes convert ATP into cAMP molecules via adenylyl cyclase III. The rate of production of cAMP is assumed independent of the available ATP (which is in excess), and is therefore proportional only to the number of activated receptors. The cAMP molecules diffuse locally and cooperatively open nearby cyclic-nucleotide-gated (CNG) channels permeable to Ca$^{2+}$ and Na$^+$ ions. Since CNG channels are distributed uniformly along the cilia (*Flannery et al., 2006*; *Takeuchi and Kurahashi, 2005*), we ignore the time required for cAMP to diffuse from its production site to the CNG channel. CNG channels have four binding sites for cAMP that exhibit allosteric cooperativity, which leads to nonlinear response functions of the Hill form (*Segel, 1993*) and constitutes the first stage of signal amplification in the ORN. The generated Ca$^{2+}$ current is then proportional to the number of fully bound CNG channels and is exchanged out of the cell at a constant rate through Na$^+$/Ca$^{2+}$ exchangers. Ca$^{2+}$ ions open Cl$^-$ channels further downstream, which produces an outward amplifying Cl$^-$ current (*Boccaccio and Menini, 2007*). Spike firing, in proportion to the current, follows. Lastly, ORN adaptation occurs due to the subsequent blocking of the CNG channels by a Ca$^{2+}$-calmodulin complex.

Despite the complexity of the transduction model, the specific identity of odorants plays a role only at the level of receptors (except for masking agents, described below). Odorants are characterized by two parameters (mathematically defined by (*Equation 15*) in the Materials and methods): the sensitivity, $\kappa^{-1} = \frac{1+\kappa_2}{\kappa_1\kappa_2}$, which controls the affinity of $X$ to the receptor, and the activation efficacy, $\eta$, which combines $\kappa_2$ with parameters of downstream reaction steps to measure the current produced by $X$ once bound. Numerically integrating the set of coupled rate equations presented in the Methods yields the temporal firing rate response of the ORN to pulses of odorant molecules and their mixtures at various concentrations (*Figure 1B–D*). For single odorants, the model successfully captures the strongly non-linear peak response for different concentrations of the odorant, the latency in response (*Rospars et al., 2003*), the quadratic rate of cAMP production (*Takeuchi and Kurahashi, 2017*), and calcium-based adaptation (*Kurahashi and Menini, 1997*) (*Figure 1E–G*). Since our focus in subsequent analysis will be on the peak response, its form is reproduced here (see Materials and methods for details) for a monomolecular odorant delivered at a concentration $C$ for a fixed, short duration:

$$F_C = \frac{F_{max}}{1 + \left(\frac{1+C/\kappa}{\eta C/\kappa}\right)^n}. \tag{2}$$

Here, $n$ is the Hill coefficient and $F_{max}$ is the maximum physiologically possible firing rate, which depends on parameters related to the transduction pathway downstream of receptor activation and can be rescaled to unity. The maximal response at saturating concentrations, $F_\infty = F_{max}/(1+\eta^{-n})$ is truncated below $F_{max}$ by $\eta$, which controls the equilibrium level of activated receptors.

On stimulation with more than one chemical species, the different species bind and activate the ORN in distinct ways. Its peak response to a pair of odorant molecules $A$ and $B$ is a special case of the general formula (*Equation 14*) derived in the Materials and methods, and reads:

$$F_{C_A + C_B} = \frac{F_{max}}{1 + \left(\frac{1 + C_A/\kappa_A + C_B/\kappa_B}{\eta_A C_A/\kappa_A + \eta_B C_B/\kappa_B}\right)^n}. \tag{3}$$

Strong amplification by the ion channels render the mixture response hyper-additive at concentrations close to the sensitivity threshold; at higher concentrations, as the receptors become saturated and the odorants compete for limited binding sites, the response turns hypo-additive. The reduction in response due to competitive antagonism is determined by the binding affinity of the weaker odorant (with lower activation efficacy) relative to the stronger one.

Before further theoretical analysis, it is worth clarifying a few points. One may question the relevance of introducing a separate $\eta$ parameter, since the saturating ORN response could instead be due to limiting factors downstream of receptor activation. Our assumption is motivated by observations made from spike recordings of single rat ORNs to odorant mixtures, where the responses of the same ORN to saturating concentrations of different odorants yield very different firing rates

(*Rospars et al., 2008*). Since odorant-ORN specific interactions occur only at the receptor, the differences at saturation must be due to differences in receptor activation efficacy. Further evidence comes from examining the minimum latency of spiking response to different odorants at saturating concentrations (*Figure 1E*), which can range from a few hundred milliseconds to a few seconds depending on the odorant-ORN pair (*Rospars et al., 2003*). The lifetime of odorants bound to a receptor is short (of a few milliseconds) and the probability of activation of a particular G-protein in a 50 ms interval at saturating concentrations is low (*Bhandawat et al., 2005*). These observations strongly suggest that the latency arises due to a relatively slow build-up of cAMP over many low-probability activation events. The differences in latency at saturating concentrations, when all receptors are bound, is therefore most likely due to differing activation probabilities, in line with our assumptions on $\eta$. Note that even though both the sensitivity $\kappa^{-1}$ and the activation efficacy $\eta$ depend on $\kappa_2$ (*Equation 15*), the low probability of activation, which is reflected in the limit $\kappa_2 \gg 1$, implies $\kappa^{-1}$ and $\eta$ depend separately on $\kappa_1$ and $\kappa_2$.

While a purely competitive model of mixture interactions captured many cases from previous experiments on ORN mixture responses, a significant fraction showed discrepancies (*Rospars et al., 2008*). Non-competitive interactions are particularly manifest in synergy or suppression, which correspond to the mixture response curve lying above or below the individual response curves for each odorant, respectively, neither of which is possible with pure competition. Below, we show how non-competitive antagonistic effects, namely masking, can generate those effects. Non-competitive inhibition due to PI3K-dependent antagonism (*Ukhanov et al., 2010*) is beyond the scope of this paper.

## Masking

Masking is the phenomenon of non-specific suppression of CNG channel currents (*Kurahashi et al., 1994*; *Takeuchi and Kurahashi, 2017*). Experimental evidence suggests that masking agents disrupt the lipid bilayer on the cell membrane, and thereby alter the binding affinity of cAMP to the CNG channels (*Takeuchi et al., 2009*). Masking agents can also be odorants (like amyl acetate), that is, they also bind to receptors and excite the transduction pathway. We suppose that the agents bind to sites on the lipid bilayer, and that multiple masking agents compete for the available sites. Similar to (*Equation 1*), the effects of a masking agent are determined by its affinity $K_M$ for the masking binding sites, and a masking coefficient $\mu$ (lying between 0 and 1), which measures its inhibitory effects once bound (see Materials and methods). The latter quantifies the lowered affinity of cAMP for the CNG channels when a masking agent is bound in the vicinity of the channel. Since the activation efficacy $\eta$ that appears in (*Equation 2*) quantifies the effective rate of signal transduction, the effect of a lowered affinity appears as a lowered value for $\eta$. Specifically, we show that $\eta \to \left(1 - \mu \tilde{M}\right)\eta$, where $\tilde{M}$ is the fraction of masking sites occupied by the masking agent. It follows that the firing rate at saturating concentrations, which depends on $\eta$, is reduced (see [*Equation 19*] in Materials and methods).

The model above (and detailed in the Materials and methods) reproduces qualitative features of odorant suppression observed in experiments (*Kurahashi et al., 1994*; *Takeuchi et al., 2013*) (*Figure 2A,B*). We stress that the presented fits serve as qualitative consistency checks; more experimental data on masking agents is required to verify the specifics of the model. Importantly, we show that both suppression and, counterintuitively, synergy are possible mixture interactions that could arise due to masking (*Figure 2C,D*), both of which have been observed in experiments with single ORNs (*Rospars et al., 2008*). To define suppression and synergy, we remark that the response curve of the mixture, taking into account competitive binding alone, always lies between the response curves of the individual components. This property is a simple consequence of the fact that the number of activated receptors for a mixture (at a particular total concentration) can never be larger/ smaller than the most/least effective odorant delivered alone at the same total concentration. Suppression can then be defined as the situation when the mixture response curve is lower than the lowest response curves among the components while synergy occurs when the mixture response is higher than the highest response curve. In our model, synergy is qualitatively due to the taming of suppressive effects: for instance, let us consider a component $A$ that binds masking sites more weakly than $B$, yet it is a stronger suppressor. If $A$ binds and activates the ORN more strongly than $B$, the net effect of mixing $A$ and $B$ is to reduce the suppressive masking effect of $A$ and unmask its strong activation properties, which can exceed the individual response curves as in *Figure 2C*.

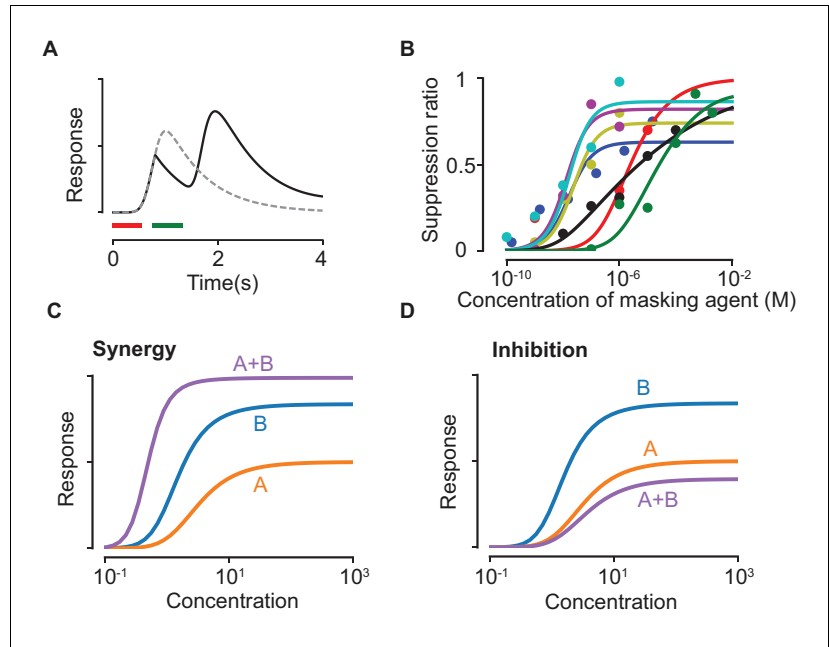

**Figure 2.** Suppression due to masking. (**A**) The dashed line shows the response predicted by our model with only the first odorant (delivered during the red time window), while the solid line shows the predicted response when a second pulse of a highly masking odorant is delivered shortly after during the green window (plotted as in (*Kurahashi et al., 1994*)). (**B**) Experimental data on the masking effect of various masking agents (circles) and fits to theory (lines). Blue: 2,4,6-trichloroanisole, red: 2,4,6-tribromoanisole, yellow: phenol, Magenta: 2,4,6-trichlorophenol, cyan: trichlorophenetole, black: L-cis diltiazem, green: geraniol. (**C–D**) Mixture response curves displaying synergy (**C**) and inhibition (**D**). The curves are plotted as in *Figure 1D*.
DOI: https://doi.org/10.7554/eLife.34958.004

## Olfactory encoding and antagonism

Equipped with the biophysical model above, we now proceed to investigate the functional consequences of antagonism. To this end, we first define a model of olfactory encoding that focuses on competitive antagonism and introduce simplifying assumptions to highlight the main ideas.

An odorant is defined by two $N$-dimensional vectors of sensitivities $\kappa^{-1}$ and activation efficacies $\eta$ across $N$ distinct ORN receptor types. We take $N = 250$, which is large enough to generalize our results across species. Parameters for different odorants are drawn independently from log-normal probability distributions (see Materials and methods), although our main conclusions below do not depend on their specific form. The width of the $\kappa^{-1}$ distribution reflects the broad sensitivities of odorants, spanning about six orders of magnitude (*Saito et al., 2009*).

An odorant caught in a sniff elicits a response in the glomerular ensemble whose individual activations vary in magnitude and progress differently in time. We focus on the vector $y$, which represents the peak responses (*Equation 2*) to the odorant for the different ORN types. The statistics of the glomerular response is encoded in the distribution of $y$, which is directly related to the distribution of $\eta$ for saturating concentrations of the odorant.

A straightforward generalization of the expression for the binary mixture response from (*Equation 3*) allows us to write the mixture response for $K$ components in terms of effective mixture parameters $\kappa_{\mathrm{mix}}$ and $\eta_{\mathrm{mix}}$ (which replace $\kappa$ and $\eta$ in [*Equation 2*]):

$$\kappa_{\mathrm{mix}}^{-1} = \sum_{i=1}^{K} \beta_i \kappa_i^{-1} \,; \, \eta_{\mathrm{mix}} = \kappa_{\mathrm{mix}} \sum_{i=1}^{K} \eta_i \beta_i \kappa_i^{-1} \tag{4}$$

where $\beta_i$ is the fraction of component $i$ in the mixture. We first consider for simplicity an equiproportionate mixture, and then show below that this is not a limitation. The key observation made here is

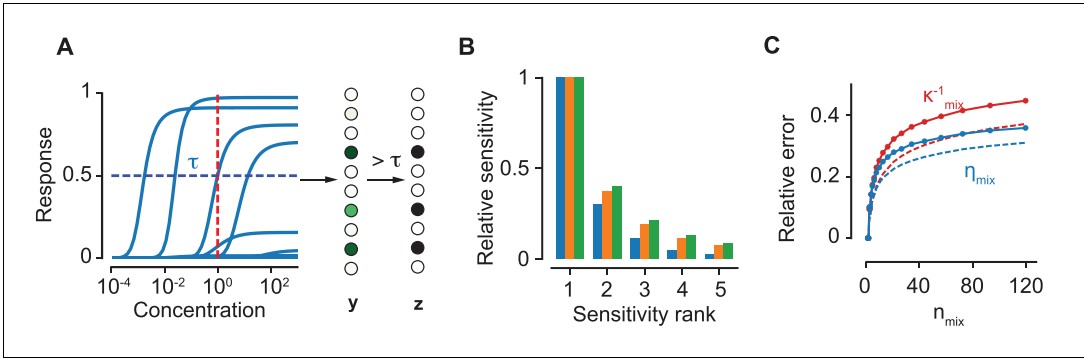

**Figure 3.** The encoding model. (**A**) The response curves of a collection of 10 ORN types to an odorant. The vector $y$ of continuous levels of response at a particular concentration (red, dashed line) yields a binary vector $z$ of activation by imposing a threshold $\tau$ (blue, dashed). (**B**) For a particular ORN type, the sensitivities of the five most sensitive odorants in a mixture relative to the sensitivity of the most sensitive odorant are shown. Blue, orange and green colors correspond to the number of components in the mixture, $n_{\mathrm{mix}} = 10$, 50 and 100 respectively. (**C**) The relative error due to the approximation in (4) for $\kappa_{\mathrm{mix}}^{-1}$ (red) and $\eta_{\mathrm{mix}}$ (blue) as a function of $n_{\mathrm{mix}}$. Solid lines refer to an equiproportionate mixture. Conversely, dashed lines refer to the case where concentrations are drawn uniformly in log scale over six orders of magnitude. The comparison indicates that our approximations in the main text become even better when the concentrations are variable.
DOI: https://doi.org/10.7554/eLife.34958.005

that, since sensitivities are broadly distributed, we typically have one term dominating the sum in the expression for $\kappa_{\mathrm{mix}}^{-1}$ (see *Figure 3B*). If we suppose that this dominant term is $\kappa_M^{-1}$, we may write:

$$\kappa_{\mathrm{mix}}^{-1} \approx \beta_M \kappa_M^{-1} \; ; \eta_{\mathrm{mix}} \approx \eta_M \,, \tag{5}$$

for typical values of $\eta_M/\eta_i$ for $i \neq M$. The expression for $\eta_{\mathrm{mix}}$ above follows from (*Equation 4*) when we approximate the sum in the expression for $\eta_{\mathrm{mix}}$ as $\eta_M \beta_M \kappa_M^{-1}$; multiplying this sum with the expression for $\kappa_{\mathrm{mix}}$ in (*Equation 5*) then yields the approximation. Relation (*Equation 5*) still holds for complex mixtures as the broad width of the sensitivity distribution ensures the dominance of one of the $\kappa$'s even for relatively large numbers of components. In *Figure 3C*, we show that the approximations in *Equation 5* result in a relative error in $\kappa_{\mathrm{mix}}^{-1}$ and $\eta_{\mathrm{mix}}$ of about $\sim 40\%$ for equiproportionate mixtures with over a hundred components. In other words, it is highly likely that only one component in the mixture occupies the receptors of ORNs of a particular type, and thus the response from these ORNs is determined by the activation efficacy of that specific component.

In order to measure the strength of competitive antagonism, we introduce the antagonistic factor $\rho$. To define $\rho$, we note that an odorant $A$ competitively antagonizes odorant $B$ (at equal concentrations) when its sensitivity exceeds that of $B$ ($\kappa_A^{-1} > \kappa_B^{-1}$), yet its activation efficacy is lower than $B$ ($\eta_A < \eta_B$). A quantification of this relationship is the Pearson correlation coefficient between binding and activation strengths across the ORN ensemble:

$$\rho \equiv \mathrm{Corr}\big(\log \kappa^{-1}, \log \eta\big), \tag{6}$$

where the logarithms conveniently account for the broad range of the two variables.

Let us first consider the extreme case of $\rho = 1$, when there is no antagonism as the odorant that binds best also has the strongest activation. This corresponds to an ORN behaving as a logical OR gate, a feature shared by any additive model of mixture response. From *Equation 5*, the consequence is that $\eta_{\mathrm{mix}}$ always takes the maximum value of $\eta$ in the mixture, which significantly biases $\eta_{\mathrm{mix}}$ toward higher values. To see this, we define the sparsity $p$, that is, the fraction of glomerular responses above a certain threshold $\tau$ for an odorant at saturating concentrations; then, the fraction above $\tau$ for a mixture with $K$ components is given by $1 - (1 - p)^K$. Thus, for any additive model of mixture response, the sparsity quickly saturates to one as $K$ increases and all information about individual components is lost.

On the contrary, when $\kappa^{-1}$ and $\eta$ are independent, that is, $\rho = 0$, $\eta_M$ is independent of the constraint $\kappa_M^{-1} \gg \kappa_i^{-1}$ ($i \neq M$) implicit in *Equation 5* and the distribution of $\eta_{\mathrm{mix}}$ precisely matches the distribution of the single component $\eta_M$. The $\rho = 0$ condition of decorrelation between $\kappa^{-1}$ and $\eta$ is important for the argument since, even though *Equation 5* holds generally, the constraint that $M$ is the most sensitive odorant biases the statistics of $\eta_{\mathrm{mix}}$ for $\rho > 0$. Since the entire distribution of activations across the ORN ensemble is invariant to the number of components in the mixture, we conclude that the statistics of activation is conserved as the complexity of the mixture increases, that is, the population response is 'normalized' (*Figure 4A–C*, *Figure 4—figure supplement 1*). Remarkably, such a normalization of the mixture response is a direct consequence of antagonism in receptor encoding, independent of any neural circuitry. The upshot is that a sparse representation for a single odorant typically remains sparse for a complex mixture, allowing for improved performance in the detection of individual components (as quantified in the next Section).

Notably, the above arguments rely solely on the broad distribution of the sensitivities, which enables the approximation in *Equation 5*. Despite the $\sim 40\%$ error in this approximation, our arguments for normalization presented above still hold for a mixture with over a hundred components, as shown in *Figure 4A*. An important consequence is that the approximation becomes even better

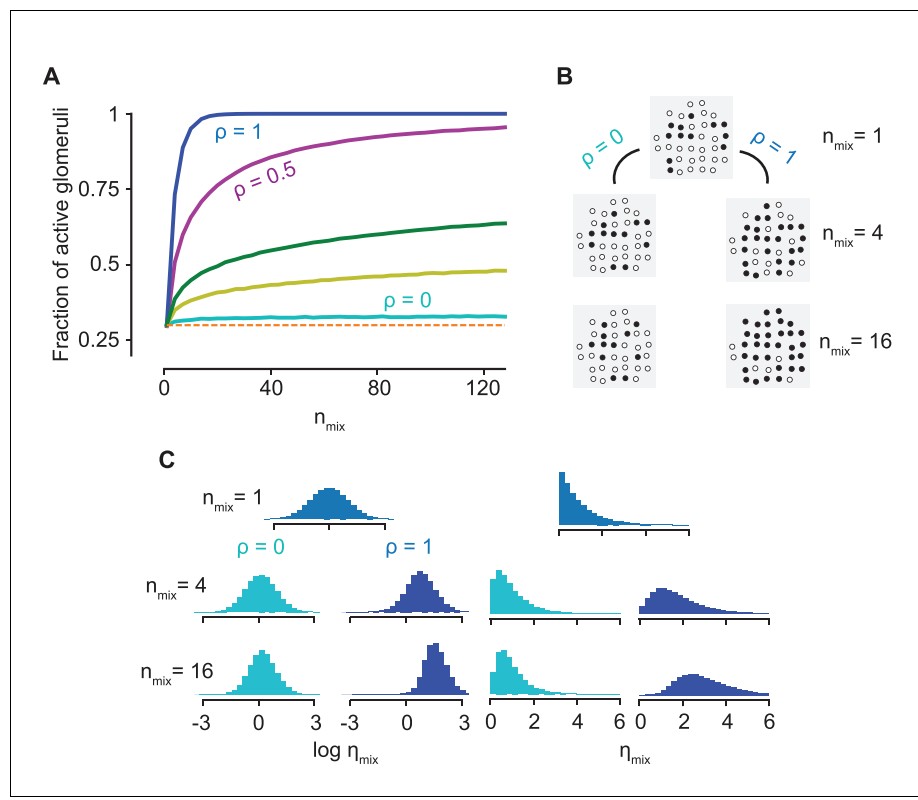

**Figure 4.** Normalization due to antagonism. (**A**) The fraction of active glomeruli as the number of odorants in the mixture, $n_{\mathrm{mix}}$, increases, each of which individually activates 30% of the glomeruli ($\rho = 0, 0.1, 0.2, 0.5, 1$ from the bottom to the top curves). (**B**) The glomerular pattern of activation for $\rho = 0$ and $\rho = 1$ for three values of $n_{\mathrm{mix}}$, shown to contrast the sparsity of their glomerular responses. (**C**) The distribution of the activation efficacy $\eta_{\mathrm{mix}}$ over different ORN types for $\rho = 0$ (cyan) and $\rho = 1$ (blue) as $n_{\mathrm{mix}}$ increases, where $\eta$ for each component is drawn from a log-normal (left) or an exponential distribution (right). The distribution is largely invariant w.r.t $n_{\mathrm{mix}}$ for $\rho = 0$, whereas it gets increasingly biased toward higher values for $\rho = 1$. Normalization is independent of the sparsity of activation (see *Figure 4—figure supplement 1*).

DOI: https://doi.org/10.7554/eLife.34958.006

The following figure supplement is available for figure 4:

**Figure supplement 1.** Normalization is independent of the sparsity of activation: The fraction of active glomeruli against the number of odors in the mixture, $n_{\mathrm{mix}}$, is shown as in *Figure 4A* for sparsity $p = 0.1$ and $p = 0.2$.

DOI: https://doi.org/10.7554/eLife.34958.007

when the concentrations of the components are allowed to be different, as any variation in the $\beta_i$'s makes the distribution of each term $\beta_i \kappa_i^{-1}$ even broader. This is confirmed by the plots in *Figure 3C*. The result holds generally true for any distribution of $\eta$ and any broad distribution of $\kappa^{-1}$; their log-normal forms are used here only to simplify subsequent calculations (*Figure 4C*).

Structural constraints at the receptor level, however, are likely to hamper a perfect decorrelation $\rho = 0$. Nevertheless, normalization does not require an exact equality and its effects fade gradually as $\rho$ increases (see *Figure 4A*). The extent of the advantageous effects of normalization (and the $\rho$ value) depends on the sparsity of activation for single odorants, the number of components in the mixture and their properties. Indeed, the effect of normalization on the detection of an odorant in the presence of a large number of other odorants depends on the balance between two opposing factors. On the one hand, normalization induces an advantageous effect of maintaining sparsity and preventing saturation of the bulb, leading to easier segmentation. On the other hand, the number of active glomeruli corresponding to each odorant is greatly reduced, which makes detection harder.

## Performance in discrimination and identification tasks

To explore how our model performs in discrimination tasks, we next compute the performance of the antagonistic encoding model described above in detecting a known odorant from a large background of unknown odorants, that is, figure-ground segregation (*Rokni et al., 2014*). The capacity of an optimal Bayesian decoder in the task depends on the mutual information $I(T; y)$ (in bits) that the glomerular pattern $y$ preserves about the presence ($T = 1$) or absence ($T = 0$) of the target. To simplify the calculation of $I(T; y)$, we convert the vector of continuous values $y$ into a binary vector $z$ by applying a threshold $\tau$ that partitions the glomeruli into two subsets, active and inactive glomeruli (see *Figure 3A*). In general, any continuous read-out is demarcated into a few discrete, distinguishable states depending on the level of intrinsic noise in the system. Taking more graded states into account will not change the qualitative result of our calculation. Specifically, as we show below, the relative performance between an antagonistic and a non-antagonistic model is still dominated by the loss of information due to glomerular saturation, which occurs independently of the number of gradations in our read-out.

*Figure 5A* demonstrates that an encoding model with significant antagonism ($\rho = 0$) contains more information than a non-antagonistic model ($\rho = 1$) as the background increases in complexity. The results are robust to the presence of significant internal variability in the transduction pathway of an ORN (*Figure 5—figure supplement 1*). When the number of odorants in the mixture is small, the glomerular pattern for the non-antagonistic model is not saturated and preserves information about the glomeruli activated by each component. For a specified sparsity $p$ (as defined in the previous section as the fraction of glomerular responses above the threshold $\tau$ at high concentrations), the non-antagonistic case is thus advantageous when the mixture complexity is less than $\sim 1/p$. However, for mixtures of higher complexity, it is useful to introduce correspondingly higher levels of antagonism in order to prevent saturation and still maintain an ability to segment out different components. To further emphasize this point, for different values of $p$, we compute the level of antagonism that maximizes information transmission when the background varies both in composition and complexity (see Materials and methods). We find that for experimentally observed levels (0.1–0.3) of sparsity (*Saito et al., 2009*; *Lin et al., 2006*; *Soucy et al., 2009*; *Vincis et al., 2012*), it is always advantageous to incorporate non-zero levels of antagonism into odorant encoding (*Figure 5B*).

We measure the performance of a linear classifier in component separation, the task of identifying several known components from a mixture, for different levels of antagonism. Component separation is qualitatively different as the information about the other known odorants can be recurrently exploited to extract more information about an odorant's presence or absence (*Grabska-Barwińska et al., 2017*). First, a linear classifier is trained to individually identify 500 known odorants in the presence of other odorants from the set. In the test phase, a mixture which contains 1 to 20 known components, uniformly chosen, is delivered. The hit rate measures the fraction of odorants that were correctly identified, while the false positives (FPs) is the number of odorants out of the 500 that were not actually present but were declared to be present. Generalized Receiver Operating Characteristics (ROC) curves are drawn by varying the detection threshold of the linear classifier for

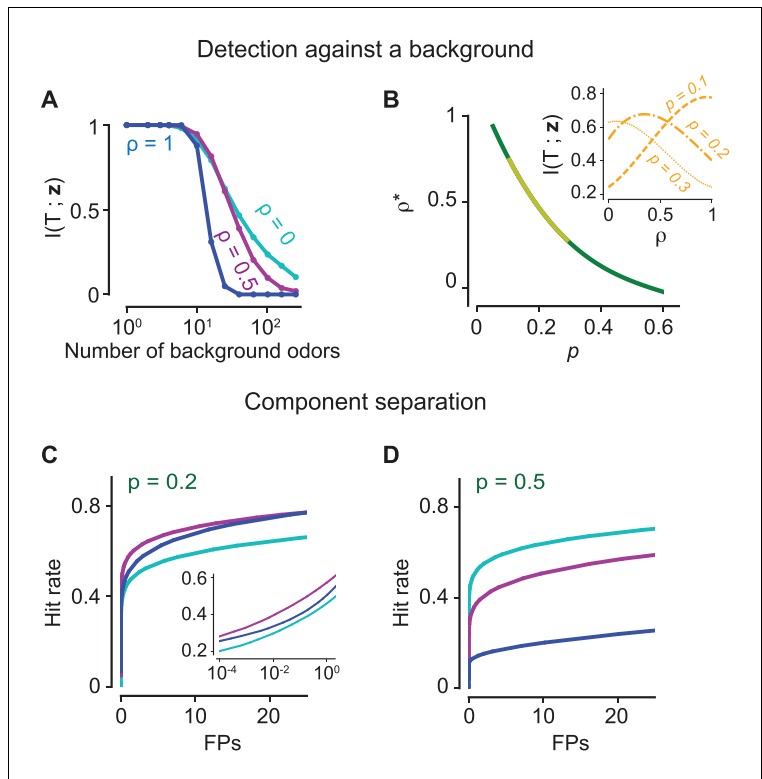

**Figure 5.** The positive effects of antagonism on odor discrimination and component separation. (**A–B**) Figure-ground segregation: (**A**) The mutual information between the presence of a target odorant and the glomerular activation vector $z$ for the antagonistic factor (13) $\rho = 0$ (cyan), 0.5 (magenta), 1 (blue) with varying number of background odorants in the mixture. (**B**) The optimal value $\rho^*$ of the antagonistic factor $\rho$ that maximizes mutual information when the number of background odorants vary from trial to trial for different sparsity levels $p$ i.e. the fraction of glomeruli that are activated. The yellow region marks experimentally observed levels of sparsity. The inset shows how the mutual information varies for three values of the sparsity. (**C–D**) Component separation: ROC curves (the hit rate vs the number of false positives (FPs)) are shown for three values of $\rho$ (color scheme as in panel A) and two relevant values of $p$. Inset: Same curves in semi-log scale. The positive effect of antagonism is retained in the case where there is significant internal noise (see *Figure 5—figure supplement 1*).

DOI: https://doi.org/10.7554/eLife.34958.008

The following figure supplement is available for figure 5:

**Figure supplement 1.** Performance is largely independent of internal noise: The accuracy of a linear classifier in detecting a target odorant is plotted against the number of background odorants for $\rho = 0$ (cyan) and $\rho = 1$ (blue), where an effective internal noise of magnitude $\epsilon$ is added (see Materials and methods).

DOI: https://doi.org/10.7554/eLife.34958.009

each odorant (*Figure 5C–D*). We find again that antagonism in receptors yields superior performance, independent of sparsity.

## Antagonism and olfactory psychophysics

Psychophysical observations related to odor perception were the primary investigative tools before neurobiological studies became prominent in the last few decades. Formulating an explicit connection between the vast body of literature on olfactory psychophysics and recent discoveries in neurobiology remains a challenge, particularly since perception is influenced by interactions throughout the olfactory sensory pathway (*Jinks and Laing, 1999*; *Su et al., 2009*; *Grossman et al., 2008*; *Wilson and Sullivan, 2011*). Various direct and indirect measurements (*Bell et al., 1987*; *Laing and Willcox, 1987*; *Chaput et al., 2012*) strongly hint at the role of receptor-level interactions, although a mechanistic explanation for how these effects may arise has not been proposed. Here, we examine the possible relation between antagonism and observations from psychophysical experiments on the

perception of odor mixtures. The upshot is that the combination of competitive antagonism and masking supports the diverse range of well-established psychophysical effects enumerated hereafter.

Specifically, the list of psychophysical effects relevant here is as follows (see (*Lawless, 1997*; *Keller and Vosshall, 2004*; *Doty and Laing, 2015*; *Thomas-Danguin et al., 2014*; *Takeuchi et al., 2013*)). (1) Inhibition and synergy: The former is the strong reduction of perceived intensities when two odorants are mixed, usually at high concentrations; synergy is occasionally observed, namely at low concentrations. (2) Masking: When the concentration of a masking agent, such as 2,4,6- trichloroanisole (called cork taint), is increased, the perceived intensity of the odorant decreases. (3) Symmetric and asymmetric suppression: When two odorants of equal perceived intensities are added, they typically suppress each other in a striking reciprocal fashion so that the perceived intensity of both odorants is still equal but sharply lowered. Asymmetric suppression (sometimes called counteracting), where the intensity of one of the odorant is lowered more than the other, is observed occasionally. (4) Overshadowing: The loss of perception of a less intense odorant when a more intense odorant is present in a mixture.

To examine the prevalence of inhibition and synergy, we estimate the inferred concentration (or perceived intensity) of a component in the mixture as the concentration at which those precise number of glomeruli corresponding to the odorant would have been activated (i.e. above a fixed threshold) had that odorant been delivered alone. In *Figure 6A*, we show that this simple algorithm leads to an unbiased estimate of the concentration over a broad range of concentrations. *Figure 6A* further demonstrates that inhibition and synergy naturally arise from competitive antagonism. Stronger inhibition arises at higher concentrations as the glomeruli that are otherwise activated by an odorant when delivered alone are antagonized by the second odorant, leading to lower perceived intensity. At lower concentrations, competitive antagonism plays a limited role; instead, cooperative effects in the ORN transduction pathway result in hyper-additivity, which pushes a few glomeruli above the activation threshold and gives rise to a small synergistic effect. Masking is also readily explained by our model, where the inferred concentration (based on the activated glomeruli) is below the actual concentration as the masking agent's concentration increases (*Figure 6B*).

To quantify suppression, we use the fraction of suppressed glomeruli, defined as the fraction of glomeruli which are inactive in the mixture of $A$ and $B$, yet are activated in isolation by odorant $A(B)$ and not activated in isolation by odorant $B(A)$. *Figure 6C* shows that the reciprocal suppression of intense binary mixtures is conspicuously absent for a non-antagonistic model. To see this, let us suppose that each odorant individually activates half of all glomeruli (i.e. the sparsity $p = 0.5$). Note, $A$ is more sensitive than $B$ to half (on average) of all the glomeruli that $B$ activates. Then, for $\rho = 0$, when they are delivered together at equal concentrations, $A$ binds better to half the otherwise active receptors of $B$, activating half of them and suppressing the other half. Since $B$ has precisely the same effect on $A$, each odorant reciprocally suppresses the other. On the other hand, when there is no antagonism, $A$ still binds better to half the active receptors of $B$, but now activates all of them, resulting in no suppression. A strong asymmetric suppressive effect is observed when one of the odorants has a capacity for masking.

Finally, to quantify overshadowing, we train a logistic regressor to identify a set of known odorants as in *Figure 5C–D*. A weak odorant $B$ is delivered along with a stronger odorant $A$ at varying concentration ratios. The probability of presence of $B$ ascomputed by the regressor is compared against the the ratio of concentrations of $A$ and $B$. When the probability of presence goes below the detection threshold (set at 0.5), $B$ is no longer detected and is 'overshadowed'. *Figure 6D* demonstrates that overshadowing for binary mixtures is intensified by antagonism, in spite of its superior discriminatory performance for more complex mixtures.

## Discussion

Natural smells are due to mixtures of many chemicals, yet the need for tight stimulus control in experiments often leads to a focus on individual molecular entities. In this paper, we have characterized mixture interactions with a realistic biophysical model. Importantly, we explored how these interactions can naturally lead to 'normalization' of the glomerular responses, improve the coding capacity of the olfactory system, and account for many observed perceptual phenomena.

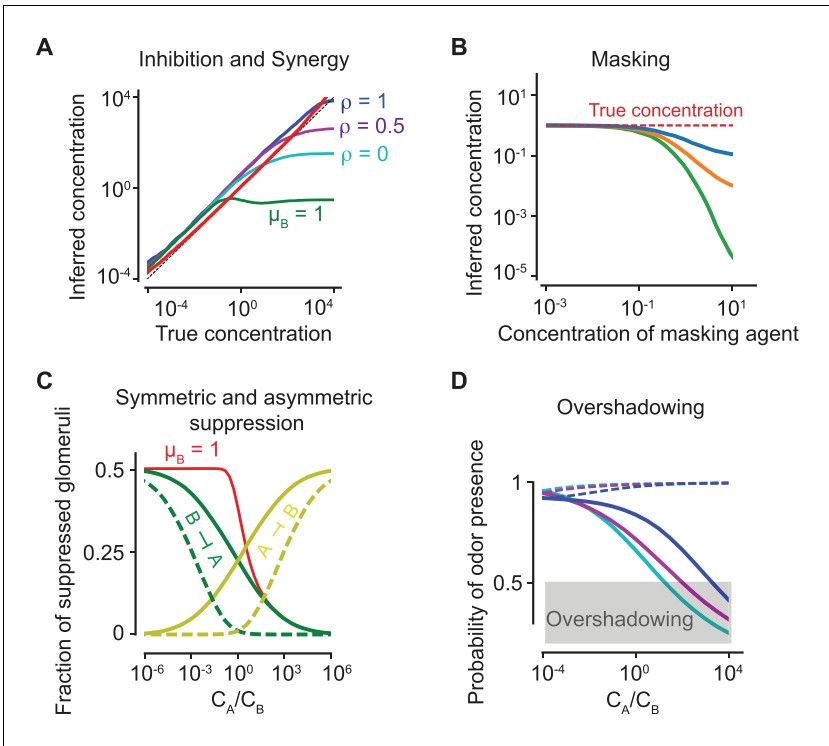

**Figure 6.** Antagonism and psychophysics. (**A**) Inhibition and synergy: The inferred *vs* the true concentration for a single odorant *A* (solid, red line) or with an additional odorant *B*. Blue, magenta, cyan: $\rho = 1, 0.5, 0$, as defined in (5). Green line: the inferred concentration when *B* has a high masking coefficient $\mu_B$, as defined in (18) and (20). In all panels, the sparsity of glomerular activation is $p = 0.5$. (**B**) Masking: The inferred concentration of *A* for increasing concentrations of a masking agent *B* (blue, orange, green: $\mu_B = 0.4, 0.7, 1$). (**C**) Symmetric and asymmetric suppression: The fraction of suppressed glomeruli of *A* (*B*) is plotted in green (yellow) against the ratio of concentrations of *A* and *B*. Solid/dashed lines: $\rho = 0, 1$. The red line shows the fraction of suppressed glomeruli when *B* also has a propensity for masking. (**D**) Overshadowing: The probability of presence of *B* as computed by the logistic regressor against the ratio of *A* and *B* concentrations. The dashed lines show the probability of presence of *A*. Color code is as in panel (**A**). The above results are independent of the sparsity of activation (see *Figure 6—figure supplement 1*).

DOI: https://doi.org/10.7554/eLife.34958.010

The following figure supplement is available for figure 6:

**Figure supplement 1.** Predicted psychophysical effects of antagonism are independent of the sparsity of activation: The results as shown in *Figure 6* for sparsity $p = 0.25$ to show that our conclusions are independent of the sparsity level.

DOI: https://doi.org/10.7554/eLife.34958.011

The odorant receptor dynamics in our model is based on a two-step activation process analogous to previous works in vertebrates (*Rospars et al., 2008*; *Cruz and Lowe, 2013*) and the fly (*Nagel and Wilson, 2011*). A key aspect of two-step activation is that it separates sensitivity of ligand binding from activation efficacy (*del Castillo and Katz, 1954*). At the structural level, this distinction is consistent with observations on the binding and activation of GPCRs (*Strange, 2008*). The common and parsimonious approximation of a single step (parametrized by a $K_d$ for affinity) in binding models, appears too drastic. In contrast to earlier work, we explicitly model the pathway downstream of receptor activation (*Pifferi et al., 2010*; *Kleene, 2008*), which features the successive steps of cAMP production, allosteric opening of CNG channels, and ultimately current fluxes. This provides a biophysical basis to the cooperativity effects that were previously introduced *ad hoc*. Moreover, this explicit formulation allows us to go beyond pure competitive antagonism, which was reported to explain about half the cases and thus requires generalizations (*Rospars et al., 2008*). In particular, non-competitive antagonistic effects, such as our masking model for the non-specific

suppression of the cyclic nucleotide-gated channels permit us to account for synergy and inhibition effects that are impossible for competitive antagonism.

A major focus of our work is the functional role of antagonistic interactions. Antagonistic reduction of glomerular activation can be seen as a form of 'normalization of activity'. Normalization with increasing stimulus intensity or complexity is common in neural systems (*Carandini and Heeger, 2011*) and has been thought of as a circuit property that involves inhibitory synaptic interactions (*Wachowiak et al., 2002*; *Cleland et al., 2007*). In the olfactory system, this was elegantly demonstrated in the *Drosophila* antennal lobe, where activation of increasing number of receptors (or glomeruli) proportionally increases inhibition provided to any one glomerular channel (*Olsen et al., 2010*). Similarly, in fish and mouse olfactory bulbs, increasing stimulus intensity is thought to recruit populations of interneurons (namely, short axon cells) that inhibit principal cells, leading to blunted activity for higher stimulus intensities. In an extreme example of this, a mouse with a particular receptor forcibly expressed widely has remarkably similar activation of the mitral cell population despite the massive increase in the input when the cognate odorant is presented (*Roland et al., 2016*).

The key insight from our model is that normalization is granted at the level of receptors by purely statistical reasons, without any additional circuit burden. In the limit of full statistical decorrelation between ligands' binding affinity and activation efficacy, the distribution of activations across the ORN ensemble for a mixture coincides with that of a single monomolecular odorant, a property which has been confirmed in the fly (*Stevens, 2016*). Why would we need normalization if the optimal way to preserve information is to simply copy the input signal, i.e. have ORNs functioning as pure relays? Copying however requires an unrealistically broad dynamic range, especially for the processing of natural mixtures, where the concentration and the number of components can fluctuate wildly. Normalization at the first layer in the sensory pathway helps avoid early saturation effects that would confound the entire processing pathway. However, nonlinear distortions of the signal do lead to loss of information, and the balance between the two effects calls for their quantification. Our information theoretic calculations demonstrate that detection of a target odorant within a complex mixture is enhanced by antagonistic interactions, and that holds for a wide range of receptor tuning widths, that is, the average number of activated glomeruli per odorant.

Some of the predictions from our analysis can be tested experimentally. Direct measurements from mammalian ORNs have been obtained in vitro in many biophysical studies studying signal transduction (*Lowe and Gold, 1995*; *Bhandawat et al., 2005*). Although such preparations offer excellent access for measurement, there is significant uncertainty about mimicking the native conditions in terms of delivery of odors (airborne vs solution) as well as the ionic composition of the perfusion medium, which will affect response amplitudes. Electrical recordings from individual ORNs are difficult and have low yield, but have offered tantalizing hints on different nonlinear interactions (*Rospars et al., 2008*). More extensive data will likely have to rely on glomerular imaging methods (*Bozza et al., 2004*; *Rokni et al., 2014*; *Mathis et al., 2016*), which offer robust signals and extended recording times to obtain measurements at different concentrations and mixture ratios (*Soucy et al., 2009*). Here too, there are some concerns including potential effects of feedback mediated by olfactory bulb neurons on ORN axon terminals, particularly through GABAb receptors (*McGann et al., 2005*; *McGann, 2013*). Carefully controlled experiments that isolate feedforward sensory signals could reveal the prevalence of antagonistic interactions in mammalian ORNs.

Extensive measurements at a large range of concentrations and mixture ratios will allow robust fitting of our model to obtain accurate estimates of the two key parameters, $\eta$ and $\kappa$. A key empirical test of our theory will then rest on the relation between $\eta$ and $\kappa$ for each glomerulus (or ORN). If these two parameters are highly correlated most of the time, then antagonistic interactions of the sort described in our theory will likely have only weak impact on olfaction. Even a modest decorrelation, on the other hand, will give rise to important effects on ensemble coding of odor mixtures even at the front end of the olfactory system.

In addition to functional advantages, we showed that antagonistic effects are consistent with psychophysical effects observed in mixture perception. Experiments show that the perceived intensity level of an odorant is empirically related to the true concentration of the odorant as a power function, reflecting their proportional relationship on a single logarithmic scale (*Lawless, 1997*; *Wojcik and Sirotin, 2014*). The intensity level for binary mixtures perception is commonly described via a vector sum of the intensities of each component (*Berglund et al., 1973*; *Laing et al., 1984*;

*Berglund and Olsson, 1993*). The vector model captures the level independent, symmetric nature of mixture suppression. The biophysical model presented here is consistent with an even broader range of perceptual phenomena, including level independency, synergy, symmetric and asymmetric suppression, masking and overshadowing. The bottomline is that global antagonistic interactions at the ORN level may play a major role in non-trivial perceptual phenomena.

In conclusion, ORNs are far from simple relays, and their strong nonlinear interactions crucially affect olfactory processing. Non-competitive antagonistic mechanisms, such as masking effects discussed here, have not been widely studied in mice and they may only occur for selected odorants. While this experimentally necessitates an extensive experimental dataset, the non-competitive effects presented here make their future investigation particularly relevant. Finally, the generality of the potential relations highlighted here between ORN antagonism and psychophysical phenomena motivates their exploration in mice, where a broader arsenal of experimental techniques and manipulations can be leveraged.

## Materials and methods

### Modeling

#### Competitive binding

When a mixture of $K$ monomolecular odorants $X_1, X_2, \ldots, X_K$ at concentrations $C_1, C_2, \ldots, C_K$ is presented to an ORN, the odorants compete for the finite number of receptors available on the olfactory cilia. In this Section, we derive the response of the ORN in such a case of competitive binding under the assumptions stated below.

The binding of an odorant to a receptor induces the activation of the odor-receptor complex, via a two-step GTP-mediated phosphorylation process (*Pifferi et al., 2010*). For a mixture of odorants, the binding dynamics reads:

$$R + X_1 \underset{}{\overset{\kappa_{1,1}}{\rightleftharpoons}} RX_1 \underset{}{\overset{\kappa_{2,1}}{\rightleftharpoons}} RX_1^*;$$
$$R + X_2 \underset{}{\overset{\kappa_{1,2}}{\rightleftharpoons}} RX_2 \underset{}{\overset{\kappa_{2,2}}{\rightleftharpoons}} RX_2^*;$$
$$\vdots \qquad \qquad \vdots$$
$$R + X_K \underset{}{\overset{\kappa_{1,K}}{\rightleftharpoons}} RX_K \underset{}{\overset{\kappa_{2,K}}{\rightleftharpoons}} RX_K^*,$$

where $RX_i$ and $RX_i^*$ symbolize the bound and activated complexes ($i = 1, 2, \ldots, K$), while $R$, $B_i$ and $A_i$ denote the number of unbound receptors, receptors bound by odorant $i$ but inactive, and receptors activated by odorant $i$, respectively. The concentration $C_i$ of the various odorants is supposed to be in excess for the total number of receptors $R_{tot} = R + \sum_{i=1}^{K} A_i + \sum_{i=1}^{K} B_i$. The parameters $\kappa_{1,i}$ and $\kappa_{2,i}$ are the ratio of the backward rates to the forward rates for the two reaction steps involving odorant $i$. We introduce only the ratio of the rates because the time scale of the slowest step, the activation of the odor-receptor complex, is estimated to be a few hundred milliseconds (*Rospars et al., 2003*). For delivery times of a few seconds or longer, we then assume equilibrium (see below). By using the steady-state relations $B_i = C_i R / \kappa_{1,i}$, $A_i = B_i / \kappa_{2,i}$, and the above equation for $R_{\text{tot}}$, we obtain that the number of activated receptors bound to odorant $i$ at equilibrium is

$$A_i^\infty = \frac{\alpha_i C_i / \kappa_i}{1 + \sum_{j=1}^{K} C_j / \kappa_j}, \tag{7}$$

with $\kappa_i = \frac{\kappa_{1,i} \kappa_{2,i}}{1 + \kappa_{2,i}}$ and $\alpha_i = \frac{R_{\text{tot}}}{1 + \kappa_{2,i}}$. Since a sniff typically lasts only for about a hundred milliseconds, the activation profile of the receptors depends on the full kinetics of sniffing, receptor binding and receptor activation. However, the number of activated receptors after a sniff of 100 ms is still proportional to those at equilibrium. Since our main conclusions below, that is, the importance of the two-step activation step and different saturation levels for different odorants, still hold, we have assumed equilibrium for simplicity and clarity of the presentation.

A chain of steps follows receptors' activation in the transduction pathway. First, activated receptors convert ATP into cAMP molecules via adenylyl cyclase III. Then, the cAMP molecules diffuse locally and open nearby cyclic-nucleotide-gated (CNG) ion channels. The open CNG channels are

permeable to $Ca^{2+}$ (and $Na^+$) ions, which are crucial in regulating further downstream processes and for adaptation (*Kurahashi and Menini, 1997*). Finally, $Ca^{2+}$ ions bind to calmodulin (CaM) and the formed complex (Ca-CaM) inhibits the CNG channels, leading to adaptation and, possibly, the termination of the response. The primary depolarizing current is carried by an $Cl^-$ efflux out of the cell through $Ca^{2+}$ regulated $Cl^-$ channels. We now proceed to model these various steps.

First, since CNG channels are spread out along the cilia membrane and cAMP diffusion is restricted to the site of its production (*Flannery et al., 2006*; *Takeuchi and Kurahashi, 2005*), successful receptor activation events are largely independent. Indeed, the electrical response is consistent with Poisson statistics, and the voltage-clamped current response close to the threshold is linear (*Bhandawat et al., 2010*). At concentrations much larger than the threshold, the production of cAMP is linearly proportional to the number of activated receptors, as evidenced by the linear increase of the rate of production of cAMP with time. The degradation of cAMP occurs on a single time scale of a few hundred milliseconds (*Takeuchi and Kurahashi, 2005*). The effective cAMP dynamics is then succinctly written as:

$$\text{RX}_i^\star \xrightarrow{k_C} \text{cAMP} \xrightarrow{d_C} \emptyset, \tag{8}$$

where $k_C$ is the rate of production of cAMP (and implicitly includes the concentration of the converted ATP, which is supposed to be in excess and thereby treated as fixed), $d_C$ is the rate of degradation, and the index $i$ runs over the set of $K$ rate equations. Since the production of cAMP occurs immediately downstream of activation and is independent of the activating odorant, the rate of production is simply proportional to the total number of activated receptors. If $C$ is the intracellular cAMP concentration, we conclude that the steady state cAMP concentration $C^\infty$ is

$$C^\infty = \frac{k_C}{d_C} \sum_{i=1}^{K} A_i^\infty. \tag{9}$$

Second, CNG channels have four binding sites for cAMP and exhibit allosteric cooperativity (*Zheng and Zagotta, 2004*), which is generally represented as (*Segel, 1993*):

$$\text{CNG}_0 + \text{cAMP} \underset{}{\overset{a_0 k_G}{\rightleftharpoons}} \text{CNG}_1 + \text{cAMP} \underset{}{\overset{a_1 k_G}{\rightleftharpoons}} \ldots \text{CNG}_{n-1} + \text{cAMP} \underset{}{\overset{a_{n-1} k_G}{\rightleftharpoons}} \text{CNG}_n. \tag{10}$$

Here, $k_G$ is an overall rate, $n$ is the number of allosteric binding sites for cAMP, and $a_0, a_1, \ldots, a_{n-1}$ modulate the various steps of the reactions. For allosteric cooperativity, $a_0 < a_1, a_2, \ldots$, which reflects the fact that the binding of one cAMP molecule promotes the binding of further cAMP molecules. Here, we have ignored the inhibitory effect of Ca-CaM, which will be introduced below. Allosteric cooperativity leads to response functions of the Hill form (*Segel, 1993*). In the limit of strong cooperativity, most of the CNG channels are expected to be either unbound or fully bound, and the steady state number of fully bound CNG channels reduces then to

$$CNG_n^\infty = \frac{CNG_{tot}}{1 + \left(k_G' C^\infty\right)^{-n}}. \tag{11}$$

Here, $C^\infty$ is the expression (*Equation 9*), $k_G = k_G \prod_{i=0}^{n-1} a_i^{1/n}$ and $CNG_{tot}$ is the total number of CNG channels. The $Ca^{2+}$ current is directly proportional to the number of fully bound CNG channels and decreases at a constant rate (*Boccaccio and Menini, 2007*):

$$\text{CNG}_n \xrightarrow{k_{ca}} \text{Ca}^{2+} \xrightarrow{d_{ca}} \emptyset \tag{12}$$

Third, the production of the CaCaM complex by $Ca^{2+}$ and CaM is described as:

$$\text{Ca}^{2+} + \text{CaM}, \overset{K_{CaCaM}}{\rightleftharpoons} \text{CaCaM}, \tag{13}$$

where $K_{CaCaM}$ is the ratio of the forward and backward rates. The effect of calmodulin-mediated feedback inhibition is accounted by assuming CaCaM modulates the CNG opening rate as $k_G' \rightarrow k_G' / \left(1 + \left(\frac{\text{CaCaM}}{\text{CaCaM}_0}\right)^2\right)$. The previous form is empirical, yet we verified that its precise shape and

the value of the Hill coefficient do not modify numerical results below as long as a steep sigmoidal shape holds. With the values of the parameters used in our model (given below), CaCaM acts on the CNG channel before the cAMP and the activated receptors reach steady state and terminates the response. The resulting set of differential equations does not lend to analytical treatment but can be numerically integrated to give a time series of the ORN response, as shown in *Figure 1B*. Numerical curves indicate (data not shown) that the $Ca^{2+}$ peak response terminated by CaM is roughly proportional to the steady state $Ca^{2+}$ response without CaM, that is, $Ca^\infty \propto \frac{k_{Ca}}{d_{Ca}} CNG_n^\infty$, which is the approximation that we shall use hereafter.

Finally, the $Cl^-$ current is the predominant component of the currents that depolarize ORNs (*Li et al., 2016*; *Boccaccio and Menini, 2007*) and is mediated by $Ca^{2+}$-gated $Cl^-$ ion channels. This current response induced due to $Ca^{2+}$ is again a Hill function of coefficient greater than one, suggesting further cooperativity (*Reisert et al., 2005*). We can formally write the steady state $Cl^-$ current as $I_{Cl}^\infty = f(Ca^\infty)$, where $f$ is some unknown function. The firing rate response $F$ of the ORN is assumed to be proportional to the current, so that $F = k_f I_{Cl}^\infty$, where $k_f$ is a constant.

By combining all the equations above, we can write the firing rate as a function of the odorant concentrations $C_i$:

$$F(C_1, C_2, \ldots, C_K) = k_f f \left( \frac{CNG_{tot}}{1 + \left( \frac{1 + \sum_{i=1}^{K} C_i / \kappa_i}{\sum_{i=1}^{K} \eta_i C_i / \kappa_i} \right)^n} \right), \tag{14}$$

where the odorant-receptor dependent parameters for the $i$th odorant are written explicitly:

$$\kappa_i = \frac{\kappa_{1,i} \kappa_{2,i}}{1 + \kappa_{2,i}}, \tag{15}$$

$$\eta_i \propto k_C k_G' \alpha_i / d_C, \quad \text{where } \alpha_i = \frac{R_{tot}}{1 + \kappa_{2,i}}.$$

As mentioned in the main text, for each odorant $i$, $\kappa_i$ and $\eta_i$ carry the dependence on the odorants and control their interactions within mixtures. Conversely, the unknown function $f$ is related to downstream processes and therefore expected to not depend on the odorant and receptor type. This point, together with the reduction in the number of free parameters, was our rationale for using the approximation of a linear function $f$. Then, the proportionality constant in $f$ is lumped together with $k_f$ and $CNG_{tot}$ in (*Equation 14*) into a single parameter $F_{max}$, which defines the maximum physiologically possible firing rate of the neuron. Both $F_{max}$ and $n$ are constants that do not depend on the odorant or the receptor type.

We can finally write a general expression for the ORN response to a mixture of $K$ odorants with concentrations $C_1, C_2, \ldots, C_K$. Denoting the total concentration by $C = \sum_i^K C_i$ and the contribution of the $i$th component by $\beta_i = C_i / C$, it follows from (*Equation 14*) that the response reads

$$F(C_{complex}) = \frac{F_{max}}{1 + \left( \frac{1 + C/\kappa_{mix}}{\eta_{mix} C / \kappa_{mix}} \right)^n}, \tag{16}$$

where the 'effective' mixture parameters $\eta_{mix}$ and $\kappa_{mix}$ are

$$1/\kappa_{mix} = \sum_{i=1}^{K} \beta_i / \kappa_i \; ; \; \eta_{mix} = \kappa_{mix} \sum_{i=1}^{K} \eta_i \beta_i / \kappa_i . \tag{17}$$

A complex odorant can therefore be treated in a manner similar to monomolecular odorants, namely, by specifying its effective sensitivity and activation efficacy to each receptor type. Note that this holds generally true, irrespective of the linear $f$ chosen in *Equation 14* to limit the number of free parameters.

The parameters used in *Figure 1* are as follows – we first define $k_1, k_{-1}$ as the forward and backward rates for the binding step of *Equation 1*, and $k_2, k_{-2}$ as the forward and backward rates for the activation step. For panels B,C and D, we use $k_1 = 100s^{-1}$, $k_{-1} = 100s^{-1}$, $k_2 = 2s^{-1}$, $k_{-2} = 2s^{-1}$ for

odorant A and $k_1 = 80s^{-1}$, $k_{-1} = 100s^{-1}$, $k_2 = 0.4s^{-1}$, $k_{-2} = 2s^{-1}$ for odorant B. The concentration is unity in panels B, F and G. For panels E, F and G, the parameters for odorant A are used. In all panels, we use $k_C = 2s^{-1}$, $d_C = 1s^{-1}$, $k_G = 10$, $CNG_{tot} = 1$, $n = 4$, $k_{Ca} = 20s^{-1}$, $d_{Ca} = 0.5s^{-1}$, $k_{CaCaM} = 1s^{-1}$, $CaCaM_0 = 0.05$.

## Masking

Here, we present a phenomenological description of non-competitive masking processes.

We suppose that masking agents bind sites on the lipid bilayer and compete for their limited number. The suppression timescale and off-timescale are smaller than a few hundred milliseconds (*Kurahashi et al., 1994*), justifying the assumption of steady state. In steady state, the occupancy fraction of the $i$th masking agent with concentration $M_i$ and binding affinity $K_{M_i}$ is

$$\tilde{M}_i = \frac{K_{M_i} M_i}{1 + \sum_i K_{M_i} M_i}. \tag{18}$$

The disruption of a CNG channel conformation due to agent $i$ is supposed to alter the affinity of cAMP to one of its binding sites on the channel in the reactions (9). The energy of the cAMP bound state is increased by $\Delta\epsilon_i$ and its probability is reduced by the corresponding Gibbs factor $e^{-\beta\Delta\epsilon_i}$, where $\beta = 1/kT$ is the inverse temperature. The resulting reduction in the opening of the channels is most conveniently accounted for by a mean-field approach where the channel opening rate $k'_G$ appearing in (10) is modified by the masking agents. In other words, $k'_G \rightarrow \chi_M k'_G$ with the suppression factor $\chi_M < 1$ derived below. It follows from the definition (13) of $\eta$ that a modification of $k'_G$ by $\chi_M$ carries over to $\eta$ as $\eta \rightarrow \chi_M \eta$. Therefore, when saturating concentrations of excitatory odorants are presented together with masking agents that produce a masking coefficient $\chi_M$, the maximal firing rate is reduced as

$$F_M(\infty) = \frac{F_{max}}{1 + 1/(\chi_M \eta)^n}. \tag{19}$$

which reflects the masking effect.

The dependence of the suppression factor $\chi_M$ on the concentrations $M_i$ of the masking agents is estimated as follows. Let us denote the radius of disruption of the channels by a single masking molecule on the lipid bilayer by $r$, and the surface density of masking binding sites by $\sigma$. The typical number of masking binding sites surrounding a given CNG channel is then $\lambda = \pi\sigma r^2$. The number $n_{mask}$ of masking binding sites within distance $r$ of a CNG channel is assumed to be Poisson distributed, that is $P(n_{mask}) = \frac{e^{-\lambda}\lambda^{n_{mask}}}{n_{mask}!}$. For a given number of sites $n_{mask}$, the vector of their occupancy numbers $I = (i_1, i_2, ..i_{K+1})$ is distributed following a multinomial distribution with probabilities given by *Equation 18*, that is, $P(I) = \binom{n_{mask}}{i_1, i_2, \ldots, i_{K+1}} \prod_{k=1}^{K+1} \tilde{M}_k^{i_k}$. The index $K + 1$ corresponds to unoccupancy, $\tilde{M}_{K+1} = 1 - \sum_{i=1}^{K} \tilde{M}_i$ and $i_{K+1} = n_{mask} - \sum_k i_k$. The probability of each $I$ is proportional to its Gibbs factor $e^{-\beta\Delta\epsilon(I)}$, where $\Delta\epsilon(I)$ is the energy shift to the binding of masking agents.

We consider the first step in *Equation 10*; similar arguments hold for successive ones. The unmodified $a_0 k_G = e^{-\beta\epsilon}$ is the ratio between the probability for a channel to be cAMP bound or cAMP unbound, and $\epsilon$ is their energy difference. In the presence of masking, there are multiple cAMP bound and unbound states, which differ in their occupancy of the masking binding sites. The sum over all those states defines the probabilities $P_b$ and $P_u$ of cAMP bound and unbound, respectively. The suppression factor $\chi_M$ that modifies $k'_G \rightarrow \chi_M k'_G$ is obtained as the ratio $(e^{\beta\epsilon} P_b / P_u)^{1/n}$, where the $1/n$ power stems from the definition of $k'_G$ in (10). The sum $P_b$ is obtained by combining all the previous factors:

$$P_b = \sum_{n_{mask}=0}^{\infty} \frac{e^{-\lambda}\lambda^{n_{mask}}}{n_{mask}!} \sum_{i_1, i_2, ..i_{K+1}} \frac{e^{-\beta(\epsilon+\Delta\epsilon(I))}}{Z} \binom{n_{mask}}{i_1, i_2, \ldots, i_{K+1}} \prod_{k=1}^{K+1} \tilde{M}_k^{i_k}. \tag{20}$$

where $Z$ is a normalization factor. Assuming the masking sites do not affect the energy of the channels when cAMP is unbound, the sum $P_u$ has a similar expression with $\epsilon + \Delta\epsilon = 0$. It is then verified

that $P_u = 1/Z$. As for $P_b$, the simplest possible assumptions are that $\Delta\epsilon(i_1, i_2, \ldots, i_K) = \sum_{k=0}^{K} i_k \Delta\epsilon_k$ is additive, and the masking binding sites are dilute, that is, $\lambda$ is small. *Equation (20)* reduces then to

$$\chi_M^n = \frac{e^{\beta\epsilon} P_b}{P_u} = e^{-\lambda \sum_k \tilde{M}_k \left(1 - e^{-\beta\Delta\epsilon_k}\right)} \approx 1 - \sum_{k=1}^{K} \mu_k \tilde{M}_k, \tag{21}$$

where $\mu_k = \lambda \left(1 - e^{-\beta\Delta\epsilon_k}\right)$ satisfy $0 \leq \mu_k \leq 1$, and the same inequality holds for $\chi_M$. In general, masking agents can affect multiple CNG channel subunits (*Chen et al., 2006*). If a masking agent affects the binding of cAMP to $j$ CNG subunits, the suppression effect is $\chi_M = \left(1 - \sum_i \mu_i \tilde{M}_i\right)^m$ with $m = j/n$.

The ratio $1 - \frac{F_M(\infty)}{F(\infty)}$ in (*Equation 19*) is plotted in *Figure 2B* and compared to experimental data. In *Figure 2C,D*, the parameters for generating the response curves for odorants A and B are $\kappa_A = 1, \kappa_B = 1, \eta_A = 1, \eta_B = 5$. For synergy (*Figure 2C*), the masking parameters are $K_{M,A} = 10^{-5}, K_{M,B} = 10^{-1}, \mu_A = 0, \mu_B = 0.7$, while the corresponding parameters for inhibition (*Figure 2D*) are $K_{M,A} = 1, K_{M,B} = 10^{-5}, \mu_A = 0, \mu_B = 0.7$. The parameter $m$ is chosen to be unity in both cases.

## Olfactory encoding model

Every odorant is defined by a vector of binding sensitivities $\kappa^{-1}$ and a vector of activation efficacies $\eta$, each with dimensionality $N$, where $N = 250$ is the chosen number of receptor types. An odorant's binding sensitivity to a particular receptor type is drawn independently from a log-normal distribution $\log\kappa^{-1} \sim N(0, \sigma_{\kappa^{-1}})$, where its standard deviation $\sigma_{\kappa^{-1}}$ is set to 4 to obtain a six orders of magnitude separation between the most sensitive and least sensitive receptor types (*Saito et al., 2009*). The activation efficacies are similarly drawn independently for each receptor type such that $\log\eta \sim N(0, 1)$. The measure of antagonism, $\rho$, is defined as the Pearson correlation coefficient between $\log\kappa^{-1}$ and $\log\eta$:

$$\rho \equiv \frac{\langle \log\kappa^{-1} \log\eta \rangle - \langle \log\kappa^{-1} \rangle \langle \log\eta \rangle}{\sigma_{\kappa^{-1}} \sigma_\eta} \tag{22}$$

where $\sigma^2$ denotes the variance of the random variables and the angular brackets denote expectation values. To generate an odorant-receptor pair, first $\log\eta$ is drawn from the standard normal distribution. Then, $\log\kappa^{-1}$ is generated with correlation $\rho$ as $\log\kappa^{-1} = \sigma_{\kappa^{-1}} \left(\rho\log\eta + \sqrt{1-\rho^2}\omega\right)$, where $\omega$ is drawn from a standard normal distribution. At saturating concentrations of an odorant, the peak firing rate it elicits for a receptor type with activation efficacy $\eta$ is $F_\infty = \frac{1}{1+\eta^{-n}}$ (see (*Equation 16*), where $F_{max}$ can be chosen to be unity). The rescaled glomerular activation vector $y$ (each component is rescaled between 0 and 1) is given by $y = \frac{1}{1+\eta^{-n}}$, where the transformation is performed on each component of the vector. The probability $p$ that each component exceed a threshold $\tau$ is given by the probability that a random variable drawn from a standard normal distribution exceed $\frac{1}{n}\log\frac{1-\tau}{\tau}$. This probability $p$ represents the sparsity of the glomerular activations $z$ at saturating concentrations after thresholding. The sparsity is set by selecting $\tau = \frac{1}{1+e^{-n\Phi^{-1}(1-p)}}$, where $\Phi$ is the cumulative distribution function for a standard normal random variable.

## Figure-ground segregation and component separation

To quantify the performance of the encoding model in figure-ground segregation, we compute the mutual information between the absence ($T = 0$) or presence ($T = 1$) of the target odorant $T$ and the glomerular activation pattern $z$. Noise is introduced due to the presence of background odorants of unknown sensitivities and activation efficacies. The mutual information controls the performance of an optimal Bayesian decoder in detecting a target odorant in a background by using the glomerular activation pattern as input. The mutual information is defined as

$$I(T; z) = H(T) - H(T|z), \tag{23}$$

where $H(T)$ is the entropy of target presence or absence, which equals one bit (since the target is present in half the trials). The second term on the right hand side $H(T|z)$ (in bits) is given by

$$H(T|z) = -\sum_z \Pr(z)\{\Pr(T=1|z)log_2\Pr(T=1|z) + \Pr(T=0|z)log_2\Pr(T=0|z)\}, \qquad (24)$$

where Bayes' formula yields $\Pr(T=1|z) = \frac{\Pr(z|T=1)}{\Pr(z|T=0)+\Pr(z|T=1)}$, and $\Pr(z) = \sum_{n_b}\Pr(z|n_b)\Pr(n_b)$, where $\Pr(n_b)$ is the distribution of the number of background odorants. $H(T|z)$ is estimated numerically by using Monte Carlo sampling. The quantities $\Pr(z|T=1)$ and $\Pr(z|T=0)$ are also computed numerically by observing that $\Pr(z|T=1) = \sum_{n_b}\Pr(z|T=1,n_b)\Pr(n_b)$. Due to the independence of the receptor types and since the background odorants are independently drawn, the probability $\Pr(z|T=1,n_b)$ factorizes into $N$ multiplicative terms, each of which can be pre-computed prior to Monte Carlo sampling. To obtain the results in *Figure 5A*, we choose $p=0.5$. When the number of background odorants fluctuate (as in *Figure 5B*), we draw $n_b$ from a truncated exponential distribution with a mean of 32 and truncated at a maximum of 128 background odorants.

To show that internal noise does not affect our results, we compare the performances of antagonistic and non-antagonistic models in figure-ground segregation by including an additional noise term in the expression for ORN response (*Equation 16*). Specifically, the response of an ORN to a mixture is modified as $\eta_{\mathrm{mix}} \to (1+\epsilon)\eta_{\mathrm{mix}}$, where $\epsilon$ is an effective noise term that condenses the variability in signal transduction relative to the number of activated receptors. We train a linear classifier to identify a target odorant against a fixed number of background odors. The target odorant and background odorants (of varying composition) are delivered at concentrations drawn from a uniform distribution in the logscale over 3 orders of magnitude. The discrimination accuracy of the linear classifier is shown in *Figure 5—figure supplement 1* for $\rho = 0,1$ and $\epsilon = 0$ and 0.4. The results show the superior performance of antagonism when the number of background odors is large even at noise levels of 40%. The higher performance for $\epsilon = 0.4$ compared to $\epsilon = 0$ for $\rho = 1$ (at >30 background odorants) occurs because the noise can desaturate glomeruli that are otherwise saturated.

For the component separation task, we use an ensemble of linear classifiers as our decoders. A linear classifier computes the probability of presence of an odorant from the glomerular activation pattern $z$ as $\frac{1}{1+exp(-(\theta.z+b))}$, where $\theta$ and $b$ are the vector of learned weights and bias, respectively. First, linear classifiers are trained to identify odorants from a fixed set $S$ of 500 odorants. During the training phase, each classifier is trained to identify the presence of its target against a background of one to ten odorants also chosen from $S$. In each trial of the test phase, 1 to 20 odorants are uniformly chosen from $S$ and the component separation performance is measured using the fraction of correct identifications (hit rate) and the number of false positives (FPs). An odorant is declared to be present if the probability of presence exceeds a detection threshold. The hit rate and the number of false positives are modulated by sliding the detection threshold of each linear classifier, yielding the generalized ROC curves in *Figure 5C,D*.

## Antagonism and olfactory psychophysics

To infer the concentration based on the glomerular profile for a single odorant, we first note that glomeruli are progressively recruited as the concentration of the odorant increases depending on their sensitivity to that odorant. Suppose a glomerulus $g_i$ corresponding to this odorant is first recruited at concentration $c_i$, where $c_i$ increases with the index $i$. Then, the contribution of $g_i$ to the total inferred log concentration, $logc$, is taken to be $logc_i - logc_{i-1}$ for $i>1$ and $logc_1$ for $i = 1$ (here $c_1$ corresponds to the concentration at which the most sensitive receptor becomes active). This simple scheme to infer the concentration is accurate for a single odorant (*Figure 6A*) and can be easily shown to have a concentration invariant Weber ratio $\sim \frac{logc_{max}/c_1}{pN}$ where $N$ is the number of receptor types, $p$ is the sparsity and $c_{max}$ is the saturating concentration (*Koulakov et al., 2007*). To obtain the results for mixtures when $\rho = 0, 0.5$ and 1, for each concentration of odorant $A$, another odorant $B$ is delivered at an equal concentration. The log concentration of $A$ is then inferred by computing the sum of the contributions from the glomeruli corresponding to odorant $A$ as described above. The curve corresponding to the masking agent is obtained similarly as above for $\rho = 0$ with $B$ having a masking coefficient $\mu_B = 1$.

To plot the curves in *Figure 6B*, odorant $A$ is delivered at a fixed concentration (red, dashed line) and a masking agent is applied at increasing concentrations (horizontal axis). Here, we use $\rho = 0$ and we assume the masking agent does not bind to anyof the receptors.

To show symmetric and asymmetric suppression in *Figure 6C*, the concentration of $A$ is fixed at a saturating concentration and the concentration of $B$ is varied to get different concentration ratios $C_A/C_B$ (horizontal axis). We define a suppressed glomerulus as one that is active when an odorant is delivered individually yet is inactive when $A$ and $B$ are delivered together. The fraction of suppressed glomeruli for each concentration ratio is averaged over many samplings of the two odorants $A$ and $B$.

To obtain the results for overshadowing, the different concentration ratios are generated similar to the method described above for *Figure 6C*. Logistic regressors that can detect odorants $A$ and $B$ are first trained to identify them when delivered alone at varying concentrations. A logistic regressor computes the probability of presence of an odorant given the glomerular pattern of activation. A detection threshold can then be applied on the probability of presence in order to declare the odorant present or absent. In *Figure 6D*, we show the probability of presence of odorant $B$ as determined by the logistic regressor corresponding to $B$ for different concentration ratios and values of $\rho$ (solid lines). Similar curves for odorant $A$ are shown as dashed lines.

## Code availability

Code for the modeling can be accessed at: https://github.com/greddy992/Odor-mixtures (*Reddy, 2018*); copy archived at https://github.com/elifesciences-publications/Odor-mixtures).

## Acknowledgements

We are grateful to JP Rospars for sharing the experimental dataset from Ref. (*Rospars et al., 2008*). We also thank Vikrant Kapoor for technical assistance and members of the Murthy Lab for helpful discussions. GR and MV were partially supported by the Simons Foundation Grant 340106. JZ and VNM were partly supported by a grant from the NIH (R01 DC014453); JZ was supported by NIH Fellowship F32 DC015938.

## Additional information

### Funding

| Funder | Grant reference number | Author |
|---|---|---|
| Simons Foundation | 340106 | Gautam Reddy<br>Massimo Vergassola |
| National Institutes of Health | R01 DC014453 | Venkatesh N Murthy |
| National Institutes of Health | F32 DC015938 | Joseph D Zak |

The funders had no role in study design, data collection and interpretation, or the decision to submit the work for publication.

### Author contributions

Gautam Reddy, Conceptualization, Formal analysis, Methodology, Writing—original draft, Writing—review and editing; Joseph D Zak, Investigation, Methodology, Writing—original draft, Writing—review and editing; Massimo Vergassola, Conceptualization, Formal analysis, Supervision, Writing—original draft, Project administration, Writing—review and editing; Venkatesh N Murthy, Conceptualization, Supervision, Writing—original draft, Project administration, Writing—review and editing

### Author ORCIDs

Gautam Reddy http://orcid.org/0000-0002-1276-9613
Joseph D Zak http://orcid.org/0000-0002-3148-5325
Venkatesh N Murthy http://orcid.org/0000-0003-2443-4252

### Decision letter and Author response

Decision letter https://doi.org/10.7554/eLife.34958.014
Author response https://doi.org/10.7554/eLife.34958.015

## Additional files

### Supplementary files
• Transparent reporting form
DOI: https://doi.org/10.7554/eLife.34958.012

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
