## [Decision Letter]

[Editors’ note: a previous version of this study was rejected after peer review, but the authors submitted for reconsideration. The first decision letter after peer review is shown below.]

Thank you for submitting your work entitled "Antagonism in olfactory receptor neurons and its implications for the perception of odor mixtures" for consideration by *eLife*. Your article has been favorably evaluated by a Senior Editor and three reviewers, one of whom, Fred Rieke, is a member of our Board of Reviewing Editors. Your article has been reviewed by 3 peer reviewers, and the evaluation has been overseen by a Reviewing Editor and a Senior Editor. The following individuals involved in review of your submission have agreed to reveal their identity: Dmitry Rinberg (Reviewer #2); Tim Holy (Reviewer #3).

Our decision has been reached after consultation between the reviewers. Based on these discussions and the individual reviews below, we regret to inform you that your work will not be considered further for publication in *eLife*, at least in present form. We would be willing to consider a revised paper as a new submission if you think you can thoroughly address the issues raised in review.

The reviewers all recognized the importance and novelty of the proposed model, but were less enthusiastic about the experimental basis of the model. A particular area of concern was the lack of any evaluation of the goodness or uniqueness of the model fits. This results in concerns about the validity of the extracted parameters. These issues are detailed in the individual reviews below. Thus, the reviewers agreed that the paper either needed stronger experimental support, or perhaps alternatively the model could be developed more fully and could form the core of a purely theoretical paper.

*Reviewer #1:*

This paper introduces a biophysical model for olfactory transduction, and then investigates the implications of this model for how mixtures of odors are encoded. I am not an expert in olfaction, so my comments are from the perspective of an interested outsider.

The paper makes a number of interesting points spanning a large range of olfactory phenomena. My primary concern is that the modeling approach does not seem sufficiently constrained by the data to draw strong conclusions. Examples of these and some smaller issues follow.

Modeling

The experimental tests of the model are somewhat limited. The model is fit to measures of response vs. concentration for individual odors and odor pairs and to masking data. Given the number of free parameters in the model, it does not seem surprising that it is able to capture these data pretty well. But it is unclear whether the model architecture is unique or if the specific fit parameters are unique. This raises concerns about drawing specific mechanistic inferences from the model. For example, a major conclusion is that normalization comes about from the two-stage receptor activation rather than inhibition within the olfactory bulb. But a model incorporating inhibition and a single-stage receptor activation might explain the data well. It is also not clear whether the model will generalize outside of the data to which it is fit, and this limits the strength of the conclusions about perceptual phenomena. A couple specific points related to this:

- The evidence for different saturation levels for different odors in Figure 3 is not great. How badly do models that do not allow for different saturation levels fit the data? If possible, higher odor concentrations would be helpful here. Could the different saturation levels reflect different numbers of ORNs rather than different receptor activation levels? Could saturation reflect inhibitory input rather than receptor properties?

- Some aspects of the data are not particularly well fit – e.g. some of the masking data and Figure 3Bi bottom right. Some discussion of those discrepancies would be helpful.

*Reviewer #2:*

In this manuscript, Reddy and colleagues investigate how antagonisms between different odors at ORNs shape odor perception using modeling and in-vivo Ca imaging. They proposed a novel computational model, which may explain multiple olfactory behavioral phenomena. The idea is novel and original. While the majority of previous works were focused on studying individual olfactory receptors and their interaction with one or a few ligands, the authors studied the statistics for multiple receptors and analyzed their role for such psychophysical phenomena as masking and figure-ground segregation.

It is very important and timely approach and I am looking forward to seeing this paper being published. However, my enthusiasm about experimental part of the paper is significantly lower.

1) The authors experimentally characterized ligand interactions for a few receptors in-vivo, on the level of glomerulus in a whole animal. This is definitely an important step forward. However, it is not clear how these experiments contribute to validation of the model. In a few presented cases, the authors demonstrated an existence of synergy and inhibition among ligands in-vivo, which may be important, but probably is incremental step compared to previous knowledge in the field. The main strength of the model is a prediction of collective phenomena for multiple receptors and psychophysical effects. No attempts were done to estimate relevant parameters from experimental data. Can, for example, an antagonism factor, ρ, be estimated based on current data?

2) The presentation of the experimental data is very limited. The authors did not provide a quantitative analysis of the goodness of the model prediction. The figures present just a few examples, without any quantitative validation. What is the confidence interval for individual fit parameters? Visual observation of the glomerulus responses (Figure 3B) make me question an accuracy of estimation of the sensitivity κ_(-1) and activation efficacy η, because of the response curves sometime do not reach their plateaus at high concentrations. How inaccuracy in estimation of these parameters affect prediction of a mixture responses. The author should provide a quantitative analysis of discrepancies between model prediction and experimentally measured mixture responses.

3) The proposed model is a very important, and the paper is lacking experimental validation, which I would not strongly insist upon. The theoretical work is valuable by itself. However, I would strongly encourage authors to have a discussion about strategies for experimental validation. (The current session "Experimental validation of the model" is misleading). What features of the model can and should be tested in the future work? The authors should provide, if possible, an estimation of necessary amount of data for validation their main conclusions.

4) Approximation in relations (6) are non-trivial. However, if there are large number of components in the mixture (β_i for the most sensitive ligand is small), k_i^(-1)* β_i for the highest affinity odorant may not be larger than the sum of the rest. The authors should discuss the limitation of this assumption.

*Reviewer #3:*

Reddy and colleagues explore olfactory coding both experimentally and theoretically. Consistent with previous studies, they find that some odorant receptors (as measured by imaging glomerular responses) appear to be poorly described by a simple single-state activation model. Experimentally using binary mixtures, they demonstrate competitive antagonism. The manuscript then turns to a theoretical analysis, demonstrating that competitive antagonism can act to normalize responses to large odorant mixtures, with advantageous consequences (in models) for foreground-background segmentation and at least qualitative alignment with previous psychophysical observations.

While antagonism has been known for many years, its potential role in normalizing odorant responses has not, to my knowledge, been noted. Previous work on this topic has explored the "obvious" mechanism for normalization, circuit inhibition. The mechanism explored here is less obvious, and its discovery counts as a genuinely creative contribution that could change how we think about important phenomena in olfaction.

However, the manuscript is not without its flaws:

- A conceptual understanding of how Figure 5C arises seems elusive. The logic and arguments surrounding Equation 6 are, in my opinion, either orthogonal or counterproductive to understanding the mechanism advanced by the authors. However, there appears to be a relatively satisfying argument based on the Holder inequality with p = 1 and q = infinity. In pseudo-LaTeX syntax, if we let s_i = C_i/κ_i, then η⋅s = \sum_i | η_i s_i| <= (\sum_i |s_i|) \max_i η_i where "max" is the infinity-norm, and thus (if we define S = \sum_i s_i) F = Fmax / (1 + ((1+S)/(η⋅s))^n) <= Fmax /(1 + ((1+S)/(S max_i η_i))^n) -> Fmax/(1 + (\max_i η_i)^(-n)) in the limit of large S. You will be closer this bound if η_i is correlated with s_i, and farther (i.e., smaller response) if it is uncorrelated. At least to this reviewer, this argument seems considerably more helpful in developing a conceptual understanding of Figure 5C.

- The experimental demonstration of competitive antagonism is a bit unsatisfying. Some of this may just be presentation (which glomeruli in Figure 3A correspond to the panels in Figure 3B? of the few glomeruli in 3A that visibly display mixture interactions, some are strikingly non-monotonic yet these do not seem to appear in 3B). Other concerns relate to the whole experimental strategy (the potential confound of presynaptic inhibition is treated too superficially, and Vucinic, Cohen, and Kosmidis should be cited in addition to McGann et al., 2005) or to the reliability of the extracted parameters (see below). Finally, one can't help but wish for some attempt to experimentally test whether the predictions of Figure 5C hold.

With regards to the "reliability of the extracted parameters" mentioned above, the main concern is whether the relatively uncorrelated nature of κ and η (Figure 3C) really reflects biology or whether this is a consequence of measurement noise. One could address this point and the previous one by showing the error bars for these fits of these parameters. From a technical standpoint, the fitting of the constants (κ and η being most important) should perhaps be viewed with some skepticism when κ gets near 1 or higher. In such circumstances, the data may not exist to accurately measure saturation, and hence the measurement of η may be unreliable. Some evidence for this can be found in Figure 3C, where there are a few "outliers" of very small η, but notably these occur pretty much only for κ near 1 or higher (log κ^(-1) near zero or lower).

[Editors’ note: what now follows is the decision letter after the authors submitted for further consideration.]

Thank you for resubmitting your work entitled "Antagonism in olfactory receptor neurons and its implications for the perception of odor mixtures" for further consideration at *eLife*. Your article has been favorably evaluated by Gary Westbrook (Senior Editor) and three reviewers, one of whom, Fred Rieke, is a member of our Board of Reviewing Editors. All of the reviewers felt the emphasis on the model and omission of the experimental data improved the paper. A few issues remain or were introduced in the revisions. These are clear in the individual reviews below. These issues need to be addressed before we can consider the paper further.

*Reviewer #1:*

This is a resubmission of a paper on olfactory coding. Experimental data has been removed and the paper has been revised to focus on the model for coding of complex odor mixtures. This is a substantial improvement as the experimental data raised a number of questions, and the model itself is an interesting and important contribution. There are several areas in which I think the paper can be strengthened; these center around making the modeling assumptions and structure clearer to a general reader:

Model description.

The main text would benefit from incorporation of some additional description of key properties of the model so that it can be read without continual reference to the Materials and methods. The description of the transduction model in the second paragraph of the subsection “Biophysics of mammalian olfactory receptor neurons” is one place that should be expanded. Similarly, the impact of masking in the subsection “Masking” should be expanded. Another example is the simplified model first introduced in the first paragraph of the subsection “Olfactory encoding and antagonism” (is there a schematic that could be added to illustrate the key points in Equation 5 – that might help quite a bit). The seventh paragraph of the aforementioned subsection is another place that could use expansion. The second paragraph of the subsection “Performance in discrimination and identification tasks” is another (give intuition for why antagonism can be bad, and point to range around 10 odors in 4A where this is true).

Discrimination analysis.

The results of the discrimination analysis are likely to depend strongly on the limiting source of noise. In the model, noise comes from randomness in the composition of the background odors. I suspect that noise originating after receptor binding, or after masking, could alter the results of this analysis. Ideally this would get explored in the model. At a minimum the paper should include some discussion of the assumption about where noise enters, and when this assumption is likely to hold.

*Reviewer #2:*

This is the second submission of the paper by Reddy et.al, about a novel mechanism for normalization in olfactory processing. The authors took away some limited experimental data, which did not add much to the paper and improved a clarity of model presentation. This is a novel and original idea, which should be published.

*Reviewer #3:*

The revised manuscript is greatly improved, particularly in the explanation of the mechanism by which decorrelated binding/efficacy generates "flat" responses as a function of number of odorants in the mixture. The removal of the experimental data is also a step forward, and I look forward to a more extensive test in a future publication. This is an important, insightful, and well-presented contribution.

---

## [Author Response]

[Editors’ note: the author responses to the first round of peer review follow.]

The reviewers all recognized the importance and novelty of the proposed model, but were less enthusiastic about the experimental basis of the model. A particular area of concern was the lack of any evaluation of the goodness or uniqueness of the model fits. This results in concerns about the validity of the extracted parameters. These issues are detailed in the individual reviews below. Thus, the reviewers agreed that the paper either needed stronger experimental support, or perhaps alternatively the model could be developed more fully and could form the core of a purely theoretical paper.

We thank the editor and the reviewers for their thoughtful comments. We have considered these options carefully and have decided that a purely theoretical paper would be most appropriate. Collecting significant amount of experimental data that excludes any postsynaptic effects, and has denser sampling of concentrations to allow robust estimates of saturation, will take significant amount of effort. Therefore, following your above suggestion, we have chosen to develop the model further and make it the exclusive focus of the paper. Briefly, the following are significant new additions:

The Introduction has been modified for improved clarity and includes an introduction to sensory coding in the early olfactory system for readers outside the field.

The section describing the biophysical model has been considerably expanded, where we justify our key assumptions in the model using evidence from previously published experimental works.

The “Olfactory encoding and antagonism” section has been significantly altered for clarity and expanded to emphasize the generality of our theoretical arguments. Importantly, we stress that antagonism can preserve the entire distribution of responses of the ORN population, regardless of the number of components in the mixture.

We have added two panels to Figure 5 (Figure 3 of the new draft) in order to justify our key assumption and lend additional support to the point noted above that the entire distribution of ORN responses is preserved.

The “Antagonism and olfactory psychophysics” section has been expanded to give intuitive explanations for the results presented in Figure 7 (Figure 5 of the new draft).

In the Discussion, we detail the kind of experimental data required to validate our theoretical arguments and discuss the challenges in obtaining extensive, high-quality data.

Reviewer #1:[…] The paper makes a number of interesting points spanning a large range of olfactory phenomena. My primary concern is that the modeling approach does not seem sufficiently constrained by the data to draw strong conclusions. Examples of these and some smaller issues follow.

In the new version of the paper, we focus on the model, and leave out the experimental data. The point of the paper is therefore, to present a theory (hypotheses and predictions), which can be tested in the future.

ModelingThe experimental tests of the model are somewhat limited. The model is fit to measures of response vs. concentration for individual odors and odor pairs and to masking data. Given the number of free parameters in the model, it does not seem surprising that it is able to capture these data pretty well. But it is unclear whether the model architecture is unique or if the specific fit parameters are unique. This raises concerns about drawing specific mechanistic inferences from the model. For example, a major conclusion is that normalization comes about from the two-stage receptor activation rather than inhibition within the olfactory bulb. But a model incorporating inhibition and a single-stage receptor activation might explain the data well. It is also not clear whether the model will generalize outside of the data to which it is fit, and this limits the strength of the conclusions about perceptual phenomena. A couple specific points related to this:- The evidence for different saturation levels for different odors in Figure 3 is not great. How badly do models that do not allow for different saturation levels fit the data? If possible, higher odor concentrations would be helpful here. Could the different saturation levels reflect different numbers of ORNs rather than different receptor activation levels? Could saturation reflect inhibitory input rather than receptor properties?- Some aspects of the data are not particularly well fit – e.g. some of the masking data and Figure 3Bi bottom right. Some discussion of those discrepancies would be helpful.

We agree with the reviewer that additional data is necessary for careful testing of the model/theory. We are actively pursuing strategies for getting additional data with greater sampling to reduce variability, and to remove any confound from postsynaptic processing, but we have chosen to focus this paper on the theory.

Reviewer #2:[…] It is very important and timely approach and I am looking forward to seeing this paper being published. However, my enthusiasm about experimental part of the paper is significantly lower.

We thank the reviewer for these supportive and critical comments. As noted above, we have chosen to follow the constructive suggestions that we received during the previous reviewing process and focus on the model/theory part, saving the experimental test for a future, more comprehensive paper. We have significantly expanded the theory part, and hope that the reviewer maintains his support for the paper.

1) The authors experimentally characterized ligand interactions for a few receptors in-vivo, on the level of glomerulus in a whole animal. This is definitely an important step forward. However, it is not clear how these experiments contribute to validation of the model. In a few presented cases, the authors demonstrated an existence of synergy and inhibition among ligands in-vivo, which may be important, but probably is incremental step compared to previous knowledge in the field. The main strength of the model is a prediction of collective phenomena for multiple receptors and psychophysical effects. No attempts were done to estimate relevant parameters from experimental data. Can, for example, an antagonism factor, ρ, be estimated based on current data?

The reviewer is correct in stating that the data currently does not allow extensive characterization of goodness of fits, robust estimates of parameters, including *ρ*.

2) The presentation of the experimental data is very limited. The authors did not provide a quantitative analysis of the goodness of the model prediction. The figures present just a few examples, without any quantitative validation. What is the confidence interval for individual fit parameters? Visual observation of the glomerulus responses (Figure 3B) make me question an accuracy of estimation of the sensitivity κ_(-1) and activation efficacy η, because of the response curves sometime do not reach their plateaus at high concentrations. How inaccuracy in estimation of these parameters affect prediction of a mixture responses. The author should provide a quantitative analysis of discrepancies between model prediction and experimentally measured mixture responses.

We have now removed the experimental data from the paper. We, however, wish to clarify some points here. Although we had presented only a few fits in the previous version, we had also presented a distribution of fitted parameters from a large number of glomerular/odor pairs (296 glomeruli). We agree that denser sampling of concentrations, especially in the saturating regime, will be better for obtaining good fits of the parameters.

3) The proposed model is a very important, and the paper is lacking experimental validation, which I would not strongly insist upon. The theoretical work is valuable by itself. However, I would strongly encourage authors to have a discussion about strategies for experimental validation. (The current session "Experimental validation of the model" is misleading). What features of the model can and should be tested in the future work? The authors should provide, if possible, an estimation of necessary amount of data for validation their main conclusions.

We thank the reviewer for the support. We have done exactly as the reviewer suggests – made clear what the experimental strategies could be for future tests of the model (Discussion, fifth and sixth paragraphs).

4) Approximation in relations (6) are non-trivial. However, if there are large number of components in the mixture (β_i for the most sensitive ligand is small), k_i^(-1)* β_i for the highest affinity odorant may not be larger than the sum of the rest. The authors should discuss the limitation of this assumption.

The approximation in relations (6) (Equation 5 of the new draft) is valid even for > 100 components in the mixture, which is now shown in a new panel in Figure 3B (of the new draft). When \β_i can vary i.e., the mixture is no longer equiproportionate, the approximation (6) in fact gets even better as the distribution of each term k_i^(-1) \β_i gets even broader. We have included and detailed this argument, as well as the corresponding results, in the new version (subsection “Olfactory encoding and antagonism”, fourth and eighth paragraphs and Figure 3B).

Reviewer #3:[…] However, the manuscript is not without its flaws:- A conceptual understanding of how Figure 5C arises seems elusive. The logic and arguments surrounding Equation 6 are, in my opinion, either orthogonal or counterproductive to understanding the mechanism advanced by the authors. However, there appears to be a relatively satisfying argument based on the Holder inequality with p = 1 and q = infinity. In pseudo-LaTeX syntax, if we let s_i = C_i/κ_i, then η⋅s = \sum_i | η_i s_i| <= (\sum_i |s_i|) \max_i η_i where "max" is the infinity-norm, and thus (if we define S = \sum_i s_i) F = Fmax / (1 + ((1+S)/(η⋅s))^n) <= Fmax /(1 + ((1+S)/(S max_i η_i))^n) -> Fmax/(1 + (\max_i η_i)^(-n)) in the limit of large S. You will be closer this bound if η_i is correlated with s_i, and farther (i.e., smaller response) if it is uncorrelated. At least to this reviewer, this argument seems considerably more helpful in developing a conceptual understanding of Figure 5C.

The arguments in the “Olfactory encoding and antagonism” section have been significantly modified to improve clarity. We note that the relation derived by using Hölder’s inequality in the above comment is indeed generally true. However, Equation 6 (of the old draft) is a much stronger statement, and arises due to the broad distribution of receptor sensitivities to different odorants. Importantly, we argue that \eta_mix, which is \sum_i \eta_i s_i (where s_i is defined in the above comment) can simply be approximated by a single term in the sum. The validity of the approximation is shown explicitly in Figure 3B of the new draft.

- The experimental demonstration of competitive antagonism is a bit unsatisfying. Some of this may just be presentation (which glomeruli in Figure 3A correspond to the panels in Figure 3B? of the few glomeruli in 3A that visibly display mixture interactions, some are strikingly non-monotonic yet these do not seem to appear in 3B). Other concerns relate to the whole experimental strategy (the potential confound of presynaptic inhibition is treated too superficially, and Vucinic, Cohen, and Kosmidis should be cited in addition to McGann et al., 2005) or to the reliability of the extracted parameters (see below). Finally, one can't help but wish for some attempt to experimentally test whether the predictions of Figure 5C hold.

We agree that much more extensive data and controls may be necessary before the glomerular imaging data become useful for testing the predictions of the model. We have, therefore, decided to focus this paper on theory and leave experimental testing to the future.

With regards to the "reliability of the extracted parameters" mentioned above, the main concern is whether the relatively uncorrelated nature of κ and η (Figure 3C) really reflects biology or whether this is a consequence of measurement noise. One could address this point and the previous one by showing the error bars for these fits of these parameters. From a technical standpoint, the fitting of the constants (κ and η being most important) should perhaps be viewed with some skepticism when κ gets near 1 or higher. In such circumstances, the data may not exist to accurately measure saturation, and hence the measurement of η may be unreliable. Some evidence for this can be found in Figure 3C, where there are a few "outliers" of very small η, but notably these occur pretty much only for κ near 1 or higher (log κ^(-1) near zero or lower).

We concede that more data are needed to obtain robust fits, and have reserved experimental tests for the future.

[Editors' note: the author responses to the re-review follow.]

Reviewer #1:This is a resubmission of a paper on olfactory coding. Experimental data has been removed and the paper has been revised to focus on the model for coding of complex odor mixtures. This is a substantial improvement as the experimental data raised a number of questions, and the model itself is an interesting and important contribution. There are several areas in which I think the paper can be strengthened; these center around making the modeling assumptions and structure clearer to a general reader:Model description.The main text would benefit from incorporation of some additional description of key properties of the model so that it can be read without continual reference to the Materials and methods. The description of the transduction model in the second paragraph of the subsection “Biophysics of mammalian olfactory receptor neurons” is one place that should be expanded.

The description of the transduction model has been considerably expanded (see subsection “Biophysics of mammalian olfactory receptor neurons”, second paragraph).

Similarly, the impact of masking in the subsection “Masking” should be expanded.

We have elaborated on the effects of masking on the mixture response and the discussion on suppression and synergy (see subsection “Masking”).

Another example is the simplified model first introduced in the first paragraph of the subsection “Olfactory encoding and antagonism” (is there a schematic that could be added to illustrate the key points in Equation 5 – that might help quite a bit).

We have added an additional panel (Figure 3B of the re-submission, note that panels from Figure 3 of the original submission have been split into Figures 3 and 4 in the re-submission) to highlight the key points in the approximation. A step-by-step derivation of Equation 5 from Equation 4 has also been added immediately after Equation 5.

The seventh paragraph of the aforementioned subsection is another place that could use expansion.

This segment has been expanded (see subsection “Olfactory encoding and antagonism”, seventh paragraph).

The second paragraph of the subsection “Performance in discrimination and identification tasks” is another (give intuition for why antagonism can be bad, and point to range around 10 odors in 4A where this is true).

We have added an intuitive explanation to delineate the regimes where an antagonistic model trumps a non-antagonistic model and vice-versa (see subsection “Performance in discrimination and identification tasks”, second paragraph).

Discrimination analysis.The results of the discrimination analysis are likely to depend strongly on the limiting source of noise. In the model, noise comes from randomness in the composition of the background odors. I suspect that noise originating after receptor binding, or after masking, could alter the results of this analysis. Ideally this would get explored in the model. At a minimum the paper should include some discussion of the assumption about where noise enters, and when this assumption is likely to hold.

We have added a supplementary figure (Figure 5—figure supplement 1 in the re-submission) showing the discrimination accuracy for the rho = 0 and rho = 1 cases with an additional internal noise term. The results show that even with 40% relative noise in transduction, the performance does not degrade significantly and the qualitative superiority of the rho = 0 case still holds. A paragraph has been added in the ‘Figure-ground segregation and discrimination tasks’ section of the Materials and methods (second paragraph) where we describe the methods for the new figure in more detail. A sentence referring to the figure has been added in the main text (see subsection “Performance in discrimination and identification tasks”, second paragraph).